# Revolutionizing Graph Aggregation:
# From Suppression to Amplification via BoostGCN

**Jiaxin Wu**
School of Management
Guangdong University of Technology, China
`704276515@qq.com`

**Chenglong Pang**
School of Computer Science and Technology
Donghua University, China
`clongdwyyx@163.com`

**Guangxiong Chen**
School of Management
Guangdong University of Technology, China
`3122003350@mail2.gdut.edu.cn`

**Jie Zhao**[*]
School of Management
Guangdong University of Technology, China
`zhaojie@gdut.edu.cn`

## Abstract

Graph Convolutional Networks (GCNs) based on linear aggregation have been widely applied across various domains due to their exceptional performance. To enhance performance, these networks often utilize the graph Laplacian norm to suppress the propagation of information from first-order neighbors. However, this approach may dilute valuable interaction information and make the model slowly learn sparse interaction relationships from neighbors, which increases training time and negatively affects performance. To address these issues, we introduce BoostGCN, a novel linear GCN model that focuses on amplifying significant interactions with first-order neighbors, which enables the model to accurately and quickly capture significant relationships. BoostGCN has relatively fixed parameters, making it user-friendly. Experiments on four real-world datasets demonstrate that BoostGCN outperforms existing state-of-the-art GCN models in both performance and efficiency.

## 1 Introduction

In recent years, Graph Convolutional Network (GCN) [1] has emerged as a powerful tool for learning from graph-structured data, with applications in various fields, such as fair recommendation [2, 3], bundle recommendation [4, 5, 6], sequential recommendation [7, 8, 9, 10] and others. The success of GCNs depends on their ability to accurately capture the user and item representation, which is achieved through an aggregation process that effectively combines collaborative signals from the nuanced interactions within the user-item graph. For example, Neural Graph Collaborative Filtering (NGCF) [11] captures user-item relationships by effectively leveraging high-order interactions through its aggregation approach. It is worth noting that some subsequent studies [12, 13] have shown that the use of feature transformations and nonlinear activation functions in aggregation can lead to significant performance degradation. Consequently, the linear aggregation-based GCN [12] has emerged as the dominant framework in this field. Then, based on the linear GCN framework, the improved methods that incorporate multimodal information [14, 15, 16, 17, 18, 19], attention mechanism [14, 20, 21], constraint loss [22, 23, 13], edge-wise dropout [24] and layer dropout [25] have blossomed.

To preserve the distinctiveness of individual nodes, existing GCN models [12, 13, 25] use the graph Laplacian norm [1] to suppress information propagation, especially from first-order neighbors. As

---

[*]Corresponding author.

39th Conference on Neural Information Processing Systems (NeurIPS 2025).

**Information Suppression**  **Information Amplification**

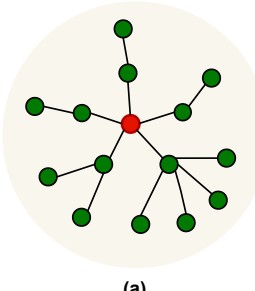 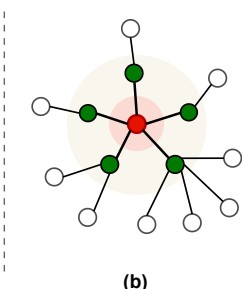

(a)                    (b)

Figure 1: Problem presentation. The red node represents the target node, while the green nodes denote its first-order or second-order neighbors. (a) shows that information propagation to the red node via the graph Laplacian norm can lead to information suppression due to considering first-order and second-order neighbors. (b) shows a new paradigm that amplifies the significant interaction information from the first-order neighbors in the information propagation.

shown in Figure 1, the red node represents the target node for information aggregation, while the green nodes represent first-order or high-order neighbors. Figure 1(a) shows the use of first-order and second-order neighbor information in GCN models, highlighting how the graph Laplacian norm stabilizes the representation scales without growth due to convolutions. While Laplacian-based models have established dominance and performance, we argue that their designs may excessively suppress the propagation of valuable information. Such models employ some stereotypical strategies that suppress items with higher interactions. However, relying solely on the number of interactions to assess the value of an item may inevitably overlook its comprehensiveness, leading to the neglect of some significant items. Specifically, the first-order neighbors, as the immediate interaction objects of the red target node in Figure 1(b), are replete with significant interaction relationships, such as conformity [26], trust [27, 28] and so on, which are particularly significant in the context of recommendation systems. The use of the graph Laplacian norm in this way can inadvertently dilute information about these significant interactions as information propagates to red nodes, which leads to a protracted and sub-optimal learning trajectory for the model.

To confirm our hypotheses, we perform an in-depth investigation of LightGCN, a quintessential representative of Laplacian-based models. As shown in Table 1, experimental results on Amazon-Book from LightGCN [12] show that the mean-based aggregation method outperforms the graph Laplacian norm-based method, which suggests that information suppression via the graph Laplacian norm may miss significant interaction information, especially for large datasets.

Table 1: Overall performance comparison between mean-based aggregation and Laplacian-based aggregation on Amazon-Book. Mean-based aggregation represents averaging the surrounding information according to the number of first-order neighbors. Laplacian-based aggregation represents a further consideration of second-order neighbors to weaken the entry of surrounding information.

| Datasets | Amazon-Book | |
|---|---|---|
| Metrics | Recall@20 | NDCG@20 |
| Laplacian-based | 0.0411 | 0.0315 |
| Mean-based | **0.0419** | **0.0320** |
| Improv.(%) | **+1.95%** | **+1.59%** |

Motivated by these empirical findings, we explore the significant interactions within the first-order neighbors, and propose a novel linear GCN model, namely BoostGCN. As shown in Figure 1(b), mining and amplifying the significant interactions in the information propagation from the first-order neighbors can be a new paradigm to improve model performance and efficiency in GCN. Accordingly, BoostGCN skilfully uses the amplification function to amplify the significant interactions in the information aggregation process, equipped with the sensitivity to quickly and accurately capture

the significant interaction cues. Moreover, this refined focus on first-order neighbor interactions is expected to result in a model that not only learns faster, but also achieves higher levels of performance.

The major contributions of this paper are listed as follows:

- We innovate graph aggregation with BoostGCN, moving from suppression to strategic amplification of significant interactions in GCNs.
- We deliberately design the amplification function of interaction significance used in Boost-GCN to be sensitive to significant interactions, thereby increasing performance and efficiency.
- Extensive experiments across four real-world datasets clearly show that our proposed BoostGCN has the best performance in terms of both recommendation performance and efficiency.

## 2  Preliminaries

For recommendations, the general GCN [12] based on linear aggregation achieves impressive performance through a straightforward process. To understand its general framework in depth, we define the interactions between users and items in a graph as: $\mathcal{G} = \{(u, i) | u \in \mathcal{U}, i \in \mathcal{I}\}$, where $\mathcal{U}$ and $\mathcal{I}$ are the user and item sets, respectively. Also, in the graph, $\mathbf{e}_u$ and $\mathbf{e}_i$ are the ID embeddings of user $u$ and item $i$. Thus, we can obtain all the original inputs:

$$\mathbf{E}^{(0)} = \{\mathbf{e}_u, \mathbf{e}_i | u \in \mathcal{U}, i \in \mathcal{I}\} \tag{1}$$

where $\mathbf{e}_u \in \mathbf{R}^{|\mathcal{U}| \times d}$ and $\mathbf{e}_i \in \mathbf{R}^{|\mathcal{I}| \times d}$; $|\mathcal{U}|$ and $|\mathcal{I}|$ are the number of users and items; $d$ is the embedding size.

Interestingly, existing linear GCNs [12, 13, 25] all leverage the graph Laplacian norm for information aggregation, as illustrated by the following formula:

$$\mathbf{e}_u^{(k+1)} = \sum_{i \in \mathcal{N}_u} \frac{1}{\sqrt{|\mathcal{N}_u|}\sqrt{|\mathcal{N}_i|}} \mathbf{e}_i^{(k)}; \mathbf{e}_i^{(k+1)} = \sum_{u \in \mathcal{N}_i} \frac{1}{\sqrt{|\mathcal{N}_i|}\sqrt{|\mathcal{N}_u|}} \mathbf{e}_u^{(k)}. \tag{2}$$

where $\mathcal{N}_u$ and $\mathcal{N}_i$ are the neighbor node sets of user $u$ and item $i$; $|\mathcal{N}_u|$ and $|\mathcal{N}_i|$ are the number of neighbors of user $u$ and item $i$; $\frac{1}{\sqrt{|\mathcal{N}_u|}\sqrt{|\mathcal{N}_i|}}$ is the graph Laplacian norm; $\mathbf{e}_u^{(k)}$ and $\mathbf{e}_i^{(k)}$ are the embeddings of user $u$ and item $i$ at layer $k$.

## 3  BoostGCN

### 3.1  Significant Interactions

In recommender systems, it is critical to accurately measure and learn user preferences from historical interactions. While it's clear that first-order neighbors with the most direct interactions have the most influence, it's significant to recognize that not all interactions are created equal. The assumption that all interactions carry the same weight contradicts intuitive understanding.

As a result, we introduce a measure of interaction significance $\mathcal{S}_{i \to u}$, which represents the significance of each item $i \in \mathcal{N}_u$ in the historical interaction relationships of user $u$:

$$\mathcal{S}_{i \to u} = f(\mathcal{X}_i) \tag{3}$$

where $\mathcal{X}_i = \{x_{i,1}, x_{i,2}, ..., x_{i,n}\}$ represents a factor set of item $i$ that contains different factors $x_{i,n}$ related to interaction significance; $f(\cdot)$ is the function that transforms these factors into interaction significance.

Drawing from previous research [27, 26, 28], we have discovered that user behavior patterns, such as conformity, trust and etc., are reflected in the interaction relationships, which are critical for mining user preferences. Consequently, they are key factors in measuring interaction significance.

Given that the factors of conformity, trust and etc., are $\{x_{i,1}, x_{i,2}, ..., x_{i,m}\}$, then we can get a factor subset $\hat{\mathcal{X}}_i = \{x_{i,1}, x_{i,2}, ..., x_{i,m}\}$ for each item $i$, where $\hat{\mathcal{X}}_i \subseteq \mathcal{X}_i$. Based on Eq.(3), we can get the

relationship between $\hat{\mathcal{X}}_i$ and $\mathcal{S}_{i \to u}$ as follows:

$$\mathcal{S}_{i \to u} = f_1(\hat{\mathcal{X}}_i) + f_2(\mathcal{X}_i \setminus \hat{\mathcal{X}}_i) \tag{4}$$

where $f_1(\cdot)$ and $f_2(\cdot)$ represent different transformation functions that are used to achieve the consistent effect of the segmented subsets and full sets of factors.

Interestingly, the effects of conformity, trust and etc., approximately match the trend of the interaction volume of items [27, 26, 28], that is, the interaction edges $|\mathcal{N}_i|$ of items $i$ in GCN. Therefore, we can always find a way that establishes the following approximate relationship:

$$f_1(\hat{\mathcal{X}}_i) = \Phi_{|\mathcal{N}_i|} + \hat{\triangle} \tag{5}$$

where $\Phi_{|\mathcal{N}_i|}$ is the estimate of $f(\hat{\mathcal{X}}_i)$ quantified by $|\mathcal{N}_i|$; $\hat{\triangle}$ is used to compensate for the estimation bias of $\Phi_{|\mathcal{N}_i|}$.

Based on Eq.(4) and Eq.(5), we can easily obtain:

$$\mathcal{S}_{i \to u} = \Phi_{|\mathcal{N}_i|} + \triangle \tag{6}$$

where $\triangle$ represents the influence of the factors in $\mathcal{X}_i \setminus \hat{\mathcal{X}}_i$, and the estimation bias compensation $\hat{\triangle}$ of $\Phi_{|\mathcal{N}_i|}$.

### 3.2 Significant Interaction Amplification

According to previous findings [27, 26, 28], we recognize that the higher the level of trust or conformity associated with an item, the greater the trend for user selection. Based on this insight, it is reasonable to assume that the more pronounced the positive effects of such factors in $\hat{\mathcal{X}}_i$, the higher the interaction significance with item $i$. Consequently, we can deduce the following relationship:

$$\mathcal{S}_{i \to u} \propto |\mathcal{N}_i| \tag{7}$$

On the basis of Eq.(7), considering the balance between model sensitivity to interaction significance and model ability to handle large $|\mathcal{N}_i|$, we choose the logarithmic function rather than the softmax method to quantify $\mathcal{S}_{i \to u}$ by $|\mathcal{N}_i|$, thereby amplifying significant interactions from neighboring items. Accordingly, Eq.(6) can be rewritten as:

$$\mathcal{S}_{i \to u}^{Amp} = \Phi_{|\mathcal{N}_i|}^{log} + \triangle^{log} \tag{8}$$

where $\Phi_{|\mathcal{N}_i|}^{log} = log_\beta(|\mathcal{N}_i|)$, $log_\beta(\cdot)$ is the amplification function and $\triangle^{log}$ denotes the offset under the logarithmic function. Based on Eq.(7), we can set $\beta > 1$ and $\triangle^{log} > 0$ to keep $\mathcal{S}_{i \to u}^{Amp} \propto |\mathcal{N}_i|$.

Given that $\varepsilon_{(i_p, i_q)}^{Amp}$ is the difference in the interaction significance of items $i_p, i_q \in \mathcal{N}_u$, then we can get:

$$\varepsilon_{(i_p, i_q)}^{Amp} = \mathcal{S}_{i_p \to u}^{Amp} - \mathcal{S}_{i_q \to u}^{Amp} = \Phi_{|\mathcal{N}_p|}^{log} - \Phi_{|\mathcal{N}_q|}^{log} \tag{9}$$

On the basis of Eq.(9), we can know that the impact of the significance estimation bias between $i_p$ and $i_q$ mainly stems from $\varepsilon_{(i_p, i_q)}^{Amp}$. The advantage of the logarithmic function is that the significance estimation bias can be adjusted directly by changing $\beta$ in the amplification function.

Therefore, for each user $u \in \mathcal{U}$ and his historical interactions with a set of items $\{i_1, i_2, ..., i_n\}$ where $i_n \in \mathcal{N}_u$, we can obtain the following amplified interaction significance for each historical interaction relationship between $u$ and $i_n$:

$$\mathcal{S}_{\mathcal{N}_u \to u}^{Amp} = \{\mathcal{S}_{i_1 \to u}^{Amp}, \mathcal{S}_{i_2 \to u}^{Amp}, ..., \mathcal{S}_{i_n \to u}^{Amp}\} \tag{10}$$

### 3.3 GCN with Information Amplification

According to the above analysis, we propose Boost Graph Convolution (BGC) method that focuses on amplifying significant interactions with first-order neighbors on the basis of Eq.(10), which allows

the model to accurately and quickly capture significant relationships. In BoostGCN, as shown in Figure 2, the graph convolution operation is:

$$\mathbf{e}_u^{(k+1)} = \sum_{i \in \mathcal{N}_u} \frac{\mathcal{S}_{i \to u}^{Amp}}{|\mathcal{N}_u|} \mathbf{e}_i^{(k)}; \quad \mathbf{e}_i^{(k+1)} = \sum_{u \in \mathcal{N}_i} \frac{\mathcal{S}_{i \to u}^{Amp}}{|\mathcal{N}_i|} \mathbf{e}_u^{(k)}. \tag{11}$$

Also, Eq.(11) can be rewritten as:

$$\mathbf{e}_u^{(k+1)} = \sum_{i \in \mathcal{N}_u} \frac{\Phi_{|\mathcal{N}_i|}^{log} + \triangle^{log}}{|\mathcal{N}_u|} \mathbf{e}_i^{(k)}; \mathbf{e}_i^{(k+1)} = \sum_{u \in \mathcal{N}_i} \frac{\Phi_{|\mathcal{N}_i|}^{log} + \triangle^{log}}{|\mathcal{N}_i|} \mathbf{e}_u^{(k)}. \tag{12}$$

where $\Phi_{|\mathcal{N}_i|}^{log}$ is calculated by the amplification function $log_\beta(|\mathcal{N}_i|)$; $\beta > 1$ and $\triangle^{log} > 0$ (we set $\triangle^{log} = 1$ in the following experiments to keep the purpose of amplification by making $\mathcal{S}_{i \to u}^{Amp} > 1$). $\mathbf{e}_u^{(k)}$ and $\mathbf{e}_i^{(k)}$ are the embeddings of user $u$ and item $i$ at layer $k$. $\mathbf{e}_u^{(0)} = \mathbf{e}_u$ and $\mathbf{e}_i^{(0)} = \mathbf{e}_i$. $|\mathcal{N}_u|$ and $|\mathcal{N}_i|$ are the number of neighbors of user $u$ and item $i$.

In BoostGCN, each $k$-th layer captures distinct information. To integrate this information effectively, we have applied a weighted sum in the combination of layers, thereby making the representation more comprehensive. After $K$ layers, we obtain the ultimate representation for a user (or an item) through a weighted combination of embeddings from each layer, which can be articulated as follows:

$$\widetilde{\mathbf{e}}_u = \sum_{k=0}^{K} \gamma_k \mathbf{e}_u^{(k)}; \quad \widetilde{\mathbf{e}}_i = \sum_{k=0}^{K} \gamma_k \mathbf{e}_i^{(k)}. \tag{13}$$

where $\gamma_k \geq 0$ denotes the significant of the $k$-th layer embedding, and we uniformly set $\gamma_k$ to $\frac{1}{K+1}$ in the following experiments.

Then, the model prediction can be defined as:

$$\widetilde{r}_{(u,i)} = \widetilde{\mathbf{e}}_u^\top \widetilde{\mathbf{e}}_i \tag{14}$$

which is used to determine the ranking score for the recommendation.

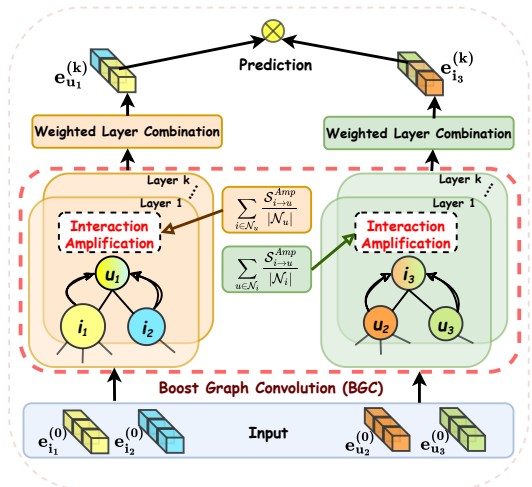

Figure 2: The overview of BoostGCN.

Following the previous research [12], we use Bayesian Personalized Ranking (BPR) loss, which is described as follows:

$$\mathcal{L}_{BPR} = \sum_{u,i,i' \in \mathcal{R}} -ln(\sigma(\widetilde{r}_{(u,i)} - \widetilde{r}_{(u,i')})) + \lambda ||\mathbf{E}^{(0)}||^2 \tag{15}$$

where $\mathcal{R} \in \{(u,i,i')|(u,i) \in \mathcal{R}^+, (u,i') \in \mathcal{R}^-\}$ is the training dataset including the observed interactions $\mathcal{R}^+$ and the unobserved interactions $\mathcal{R}^-$; $\sigma(\cdot)$ and $\lambda$ denote the sigmod function and regularization weight, respectively; the trainable parameters of BoostGCN are only the initial embeddings of the 0-th layer of users and items, i.e., $\Theta = \{\mathbf{E}^{(0)}\}$, where $||\mathbf{E}^{(0)}||^2 = \sum_{u \in \mathcal{U}} ||\mathbf{e}_u^{(0)}||^2 + \sum_{i \in \mathcal{I}} ||\mathbf{e}_i^{(0)}||^2$.

## 4  Theoretical Analysis of BoostGCN

**The selection criteria for the amplification function.** To balance the signal gain, robustness and learnability of significant interactions, we propose the following three criteria for selecting a node amplification function: (i) it should increase monotonically with the number of interactions so that nodes with a high number of interactions contribute more information; (ii) it should exhibit non-linear growth to prevent over-amplifying extremely interactive nodes and introducing noise; and (iii) it should be a closed-form, differentiable expression for efficient gradient-based optimization.

Consequently, the logarithmic amplification function is the optimal closed-form solution under these criteria. Comparative analysis with other amplification functions is provided in Appendix A.2.

**Advantages of the logarithmic amplification function.** The logarithmic amplifier in Eq.(8) exposes a single scalar $\beta$ that instantiates a tunable trade-off between magnitude amplification and model sensitivity. A larger $\beta$ compresses inter-item differences $\varepsilon^{Amp}_{(i_p,i_q)}$, yielding a more uniform signal landscape, whereas a smaller $\beta$ stretches these gaps, heightening sensitivity to subtle interaction sparsity. Detailed derivations and analysis are provided in Appendix A.2.

While Appendix A.3 details the advantages of BoostGCN's aggregation over the aggregation using graph attention, we here focus on its general-purpose properties as a universal aggregation paradigm.

**Proposition 4.1** *(Smaller aggregation error) With BoostGCN's aggregation in Eq.(12), the error bound decays at the rate of $O(log_\beta(|\mathcal{N}_i|)+1)$ as the neighbor cardinality $|\mathcal{N}_i|$ increases, guaranteeing diminishing aggregation error even for highly interactive nodes.*

**Proposition 4.2** *(Near-optimal provable linear convergence rate) By using the logarithmic amplifier in Eq.(8), BoostGCN can achieve a spectral-radius bound arbitrarily close to the infimum of the sub-linear family and yield a near-optimal provable linear convergence rate.*

Next, we give the stochastic convergence bounds of BoostGCN under stochastic training conditions.

**Theorem 4.3** *Under independent and identically distributed (i.i.d.) mini-batch sampling, bounded gradients and L-smoothness, BoostGCN's expected error bound is*

$$\mathbb{E}[||H^{(K)} - H^{tru}||_F^2] \leq (1 - 2\eta\lambda)^K \Delta_0 + \eta G^2/(2\lambda) \tag{16}$$

*And BoostGCN's high-probability bound is*

$$||H^{(K)} - H^{tru}||_F = O((log_\beta(max_j|\mathcal{N}_j|)+1)^K + \sqrt{\frac{log\frac{1}{\delta}}{K}}), with\ probability\ at\ least\ 1-\delta. \tag{17}$$

*where $H^{(K)}$ is the current embedding matrix of all users and items after $K$ rounds update of random training and $H^{tru}$ is the ideal embedding when the loss function converges to the minimum value, where $H^{(K)}, H^{tru} \in \mathbf{R}^{(|\mathcal{U}|+|\mathcal{I}|)\times d}$; $\eta$ is the learning rate and $\lambda$ is the regularization term; $G$ is the boundary of gradients, where $G > 0$ and $||\nabla\mathcal{L}|| \leq G$ for any mini-batch; $\Delta_0 = \mathbb{E}[||H^{(0)} - H^{tru}||_F^2]$ and $\delta \in (0,1)$ is confidence parameter. The proofs of the above-mentioned properties can be referred to Appendix A.4.*

# 5 Experiments

In our experiments, the following questions will be answered:

- **RQ1**: How does the performance of BoostGCN compare with state-of-the-art GCN models?
- **RQ2**: How efficient is BoostGCN?
- **RQ3**: How do different hyper-parameter settings affect BoostGCN?
- **RQ4**: What benefits does the amplification of significant interactions offer to the representation?
- **RQ5**: How does BoostGCN perform in terms of popularity debiasing and noise resistance?

## 5.1 Experimental Settings

**Datasets:** To comprehensively demonstrate the effectiveness of BoostGCN, we evaluate our model on four distinct datasets, including MovieLens-100k (denoted by 100k) [29], MovieLens-1M (denoted by 1M) [29], Gowalla (denoted by Gowa.) [30] and Yelp2018 (denoted by Yelp) [11], as detailed in Table 2. The datasets utilized in this paper are all publicly available and can be directly downloaded from their respective sources. Detailed dataset descriptions are given in Appendix A.6.

Table 3: The comparison of overall performance on four real-world datasets. **Bold** means the optimal performance, and underline means the sub-optimal performance.

| Dataset | Metric | MF-BPR | MMGCN$^{id}$ | NGCF | UltraGCN | IMP-GCN | NSE-GCN | LayerGCN | LightGCN | LTGNN | TransGNN | GAT | BoostGCN |
|---|---|---|---|---|---|---|---|---|---|---|---|---|---|
| 100k | R@5 | 0.2636 | 0.2463 | 0.2318 | 0.2647 | 0.2864 | 0.2963 | 0.2851 | 0.2861 | 0.2833 | 0.2733 | 0.2879 | **0.2982** |
| | N@5 | 0.6599 | 0.6181 | 0.6067 | 0.6575 | 0.7029 | 0.7209 | 0.6840 | 0.6971 | 0.6872 | 0.6630 | 0.6983 | **0.7234** |
| | R@15 | 0.5289 | 0.4953 | 0.4929 | 0.5393 | 0.5611 | 0.5886 | 0.5563 | 0.5669 | 0.5612 | 0.5415 | 0.5703 | **0.5908** |
| | N@15 | 0.6616 | 0.6190 | 0.6126 | 0.6662 | 0.7028 | 0.7301 | 0.6894 | 0.7044 | 0.6952 | 0.6708 | 0.7065 | **0.7319** |
| 1M | R@5 | 0.2165 | 0.2178 | 0.2081 | 0.2389 | 0.2424 | 0.2546 | 0.2599 | 0.2476 | 0.2370 | 0.2581 | 0.2537 | **0.2641** |
| | N@5 | 0.7220 | 0.7183 | 0.7128 | 0.7554 | 0.7624 | 0.7863 | 0.7949 | 0.7770 | 0.7235 | 0.7881 | 0.7844 | **0.8063** |
| | R@15 | 0.4609 | 0.4648 | 0.4545 | 0.5057 | 0.4965 | 0.5128 | 0.5251 | 0.5058 | 0.4770 | 0.5196 | 0.5106 | **0.5316** |
| | N@15 | 0.7041 | 0.7011 | 0.6934 | 0.7464 | 0.7441 | 0.7651 | 0.7782 | 0.7571 | 0.7075 | 0.7707 | 0.7574 | **0.7885** |
| Gowa. | R@5 | 0.1996 | 0.0706 | 0.1485 | 0.2288 | 0.2808 | 0.2946 | 0.2932 | 0.2521 | 0.2178 | 0.2913 | 0.2883 | **0.2965** |
| | N@5 | 0.2836 | 0.1122 | 0.1892 | 0.3131 | 0.3733 | 0.3906 | 0.3890 | 0.3408 | 0.2893 | 0.3863 | 0.3828 | **0.3937** |
| | R@15 | 0.3447 | 0.1400 | 0.3310 | 0.3851 | 0.4868 | 0.5123 | 0.5092 | 0.4476 | 0.3780 | 0.5055 | 0.5003 | **0.5145** |
| | N@15 | 0.3125 | 0.1251 | 0.2506 | 0.3460 | 0.4235 | 0.4468 | 0.4444 | 0.3883 | 0.3298 | 0.4410 | 0.4365 | **0.4489** |
| Yelp | R@5 | 0.1225 | 0.0826 | 0.0984 | 0.1435 | 0.1753 | 0.1677 | 0.1845 | 0.1545 | 0.1375 | 0.1640 | 0.1635 | **0.2001** |
| | N@5 | 0.2225 | 0.1582 | 0.1747 | 0.2581 | 0.3110 | 0.2945 | 0.3238 | 0.2747 | 0.2416 | 0.2883 | 0.2874 | **0.3517** |
| | R@15 | 0.2592 | 0.1900 | 0.2460 | 0.3021 | 0.3567 | 0.3478 | 0.3722 | 0.3213 | 0.2726 | 0.3253 | 0.3243 | **0.3968** |
| | N@15 | 0.2498 | 0.1819 | 0.2186 | 0.2901 | 0.3455 | 0.3324 | 0.3611 | 0.3086 | 0.2658 | 0.3172 | 0.3162 | **0.3869** |

**Data Pre-processing:** Each dataset is divided into training (60%), validation (20%), and test sets (20%). The training set maintains a 1:1 ratio of positive to negative samples. For MovieLens-100k and MovieLens-1M datasets, we add 150 unused negative samples for each user to the validation and test sets, respectively. Correspondingly, for Gowalla and Yelp2018 datasets, we increase the number to 1500 negative samples. The negative samples of the training set, validation set and test set are all randomly selected without popular bias.

Table 2: The description of datasets.

| Datasets | #Interactions | #Users | #Items | Sparsity |
|---|---|---|---|---|
| **MovieLens-100k** | 91,076 | 943 | 1,286 | 92.49% |
| **MovieLens-1M** | 898,224 | 6,036 | 2,864 | 94.80% |
| **Gowalla** | 1,027,370 | 29,858 | 40,981 | 99.92% |
| **Yelp2018** | 1,561,406 | 31,668 | 38,048 | 99.87% |

**Evaluation Metrics:** We employ Recall (R) and Normalized Discounted Cumulative Gain (N) [31] as the main performance metrics for Top-$N$ recommendations [32], where $N = \{5, 15\}$. For the sake of convergence, the early stop and total epochs are set to 10 and 1000, respectively. We run each experiment ten times and report the average results.

**Baselines:** We compare some representative models, ranging from traditional matrix factorization models to state-of-the-art GCN-based models: **MF-BPR** [33], **MMGCN** [29], **NGCF** [11], **Light-GCN** [12], **UltraGCN** [13], **IMP-GCN** [34], **NSE-GCN** [35], **LayerGCN** [25], **LTGNN** [36], **TransGNN** [37] and **GAT-LightGCN** (combining LightGCN with Graph Attention Network). In order to make a fair comparison, we carefully tune the hyper-parameters of each model based on their respective published papers and hyper-parameter studies. All baselines compared in this paper can be obtained directly from the corresponding literature. Detailed baseline descriptions are given in Appendix A.5.

**Implementation Details:** To ensure a fair comparison, we set the embedding size to 64, a learning rate of $10^{-3}$ with Adam [38] and initialize the embedding parameters with the Xavier method [39], which is the same as other baselines. To show the high generalization of BoostGCN on different datasets without parameter tuning, we set the fixed parameters: a batch size of 512, $\beta = \{e\}$ and $\triangle^{log} = 1$ in Eq.(12). Also, we provide guidance on $\lambda$ for different types of datasets to optimize model performance. We employ $\lambda \leq \{1e^{-6}\}$ for datasets of #Interactions $\geq 1 \times 10^6$, $\lambda = \{1e^{-4}\}$ for datasets of #Interactions $\leq 5 \times 10^5$ and $\lambda = \{1e^{-5}\}$ for others. Our code can be available on https://github.com/ClongPang/BoostGCN.

## 5.2 Experimental Results

**Performance Comparison (RQ1):** Due to the excellent performance of most baselines at layer 2, we set $K = 2$ to evaluate overall performance for a fair comparison, as shown in Table 3. From Table 3, we can easily see that: BoostGCN maintains the best performance against all baselines across four datasets of different sparsity and data size, which shows that it has stable performance in handling different types of datasets.

Table 4: The comparison of performance between BoostGCN and LightGCN at different layers. **Bold** indicates the optimal performance at each layer on four datasets. * represents the best performance of different layers on the target dataset. Improv. denotes the improvement of BoostGCN against LightGCN at each layer.

| | Dataset | 100k | | 1M | | Yelp | | Gowa. | |
|---|---|---|---|---|---|---|---|---|---|
| #Layer | Model | R@5 | N@5 | R@5 | N@5 | R@5 | N@5 | R@5 | N@5 |
| 1 | LightGCN | 0.2840 | 0.6937 | 0.2413 | 0.7647 | 0.1630 | 0.2895 | 0.2614 | 0.3517 |
| | BoostGCN | **0.2946** | **0.7157** | **0.2607** | **0.7997** | **0.1824** | **0.3258** | **0.2746** | **0.3631** |
| | Improv. | +3.73% | +3.17% | +8.04% | +4.58% | +11.90% | +12.54% | +5.05% | +3.24% |
| 2 | LightGCN | 0.2861 | 0.6971 | 0.2476 | 0.7770 | 0.1545 | 0.2747 | 0.2521 | 0.3408 |
| | BoostGCN | **0.2982*** | **0.7234*** | **0.2641*** | **0.8063*** | **0.2001** | **0.3517** | **0.2965** | **0.3937** |
| | Improv. | +4.23% | +3.77% | +6.66% | +3.77% | +29.51% | +28.03% | +17.61% | +15.52% |
| 3 | LightGCN | 0.2727 | 0.6719 | 0.2467 | 0.7763 | 0.1461 | 0.2598 | 0.2468 | 0.3329 |
| | BoostGCN | **0.2894** | **0.7075** | **0.2601** | **0.7959** | **0.1991** | **0.3493** | **0.3030** | **0.4031** |
| | Improv. | +6.12% | +5.30% | +5.43% | +2.51% | +36.28% | +34.45% | +22.77% | +21.09% |
| 4 | LightGCN | 0.2021 | 0.5339 | **0.2405** | **0.7643** | 0.1396 | 0.2494 | 0.2294 | 0.3138 |
| | BoostGCN | **0.2973** | **0.7134** | 0.2373 | 0.7556 | **0.2026*** | **0.3547*** | **0.3165*** | **0.4151*** |
| | Improv. | +47.11% | +33.62% | -1.33% | -1.14% | +45.13% | +42.22% | +37.97% | +32.28% |

Table 5: The comparison of efficiency on various datasets. Experiments are tested on the same Intel(R) Xeon(R) Platinum 8255C CPU @2.50GHz machine with a GeForce RTX 2080Ti GPU. Improv. in training time refers to Eq.(18).

| Dataset | Model | Best R@5 | Training Time |
|---|---|---|---|
| 100k (K=2) | LightGCN | 0.2861 | 45s |
| | BoostGCN | **0.2982** | **23s** |
| | Improv. | **+4.23%** | **+49%** |
| 1M (K=2) | LightGCN | 0.2476 | 1174s |
| | BoostGCN | **0.2641** | **440s** |
| | Improv. | **+6.66%** | **+63%** |
| Gowa. (K=4) | LightGCN | 0.2294 | 2762s |
| | BoostGCN | **0.3165** | **859s** |
| | Improv. | **+37.97%** | **+69%** |
| Yelp (K=4) | LightGCN | 0.1396 | 4525s |
| | BoostGCN | **0.2026** | **438s** |
| | Improv. | **+45.13%** | **+90%** |

Next, we compare our proposed BoostGCN with the basic framework LightGCN at different layers on four datasets, the results of which are shown in Table 4. We have the following observations: 1) BoostGCN outperforms LightGCN on almost all layers of four datasets, with the exception of only one case. 2) For relatively small datasets ($\#\text{Interactions} < 1 \times 10^6$) such as 100k and 1M, BoostGCN achieves optimal performance at the second layer. This could be due to the fact that these datasets, despite their smaller size, have a sufficiently dense user interaction profile, allowing the model to effectively capture the necessary interaction information within a limited number of layers. 3) For large datasets ($\#\text{Interactions} \geq 1 \times 10^6$) such as Gowalla and Yelp2018, BoostGCN consistently outperforms LightGCN across all four layers, with the most notable performance achieved at the fourth layer. This performance demonstrates BoostGCN's ability to effectively penetrate deeper into the network layers.

**Efficiency Comparison (RQ2):** To thoroughly show the advantages of BoostGCN, we perform an efficiency comparison between BoostGCN and LightGCN on different datasets. Our performance analysis, as detailed in Table 4, allows us to identify the optimal layer configuration for BoostGCN on each dataset. For the relative improvement of time efficiency in Table 5, we employ the following formula:

$$\text{Improv.} = \frac{\text{Long} - \text{Short}}{\text{Long}} \times 100\% \tag{18}$$

where "Long" and "Short" represent the long and short training time of the two compared models, respectively.

We present the results in Table 5 and it is easy to find that: 1) BoostGCN not only demonstrates superior efficiency across all datasets, but also maintains a significant improvement in performance.

2) Across the four datasets, BoostGCN records a minimum efficiency improvement of nearly 50%. More importantly, on larger dataset such as Yelp2018, BoostGCN achieves a significant leap with an efficiency improvement of 90% and a performance improvement of 45.13%.

## 5.3 Parameter Analysis (RQ3)

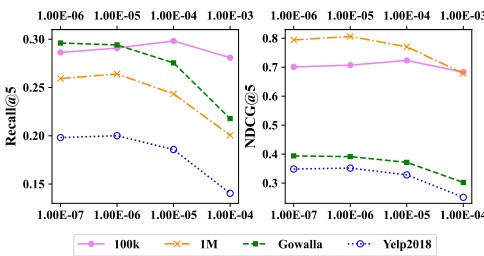

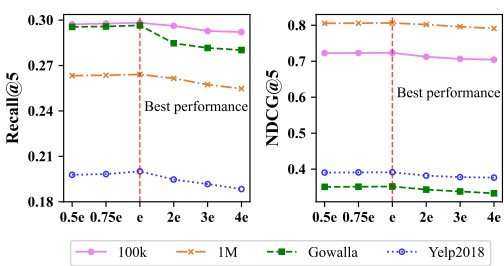

Figure 3: The performance of BoostGCN at $K = 2$ with different regularization weights $\lambda$ on four datasets. For 100k and 1M datasets, we explore the performance variation with $\lambda = \{1e^{-6}, ..., 1e^{-3}\}$. For Yelp2018 and Gowalla datasets, we explore the performance variation in $\{1e^{-7}, ..., 1e^{-4}\}$.

Figure 4: The performance of BoostGCN at $K = 2$ via different $\beta$ in Eq.(8) on four datasets. The parameter $\beta$ ranges from $\{0.5e, 0.75e, e, 2e, 3e, 4e\}$. On these four datasets, the optimal performance of BoostGCN is when the parameter $\beta = \{e\}$.

**Effect of Parameter $\lambda$:** Figure 3 vividly illustrates the impact of the regularization weight on the performance of BoostGCN across various datasets, and there are three key observations: 1) On all datasets, BoostGCN shows stable performance within $\lambda = \{1e^{-7}, 1e^{-6}, 1e^{-5}\}$, which indicates that BoostGCN is robust to parameter variations. 2) For relatively large datasets such as 1M, Gowalla and Yelp2018, a smaller $\lambda$ is more beneficial for performance improvement. 3) Our analysis yields the following guideline for parameter $\lambda$ for BoostGCN: $\lambda \leq \{1e^{-6}\}$ if #Interactions $\geq 1 \times 10^6$.

**Effect of Parameter $\beta$:** To demonstrate the simplicity of our model and the robustness to parameters, we conduct experiments on the parameter $\beta$ of amplification function in Eq.(8), as detailed in Figure 4, and have the following results: 1) BoostGCN achieves optimal performance with $\beta = \{e\}$ on all datasets when the parameter $\beta$ ranges from $\{0.5e, 0.75e, e, 2e, 3e, 4e\}$. 2) According to Eq.(8), we can easily know that with the decrease of $\beta$, the interaction significant $\mathcal{S}_{i \to u}^{Amp}$ of each item $i$ is amplified more and the model tends to be improved more. 3) This result validates the correctness of focusing on significant interactions and shows the effectiveness and superiority of information amplification in BoostGCN.

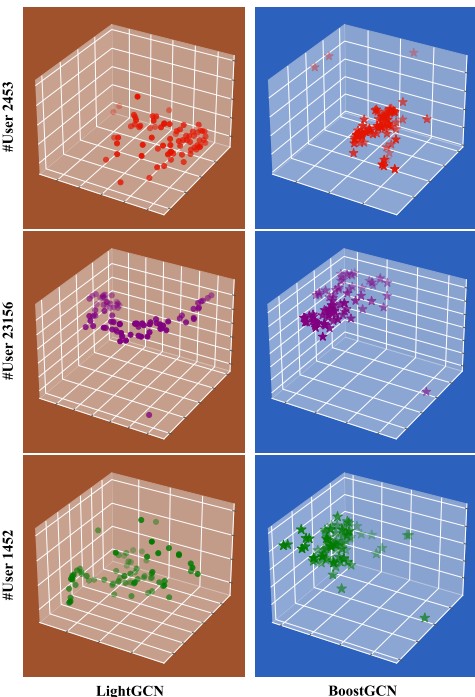

Figure 5: The visualization comparison between BoostGCN and LightGCN at $K = 4$ on Yelp2018 dataset. Red, purple, and green represent positive samples of three different users in the test set.

## 5.4 Visualization Analysis (RQ4)

To further demonstrate the benefits of amplifying significant interactions on model performance, we conduct a visualization analysis on Yelp2018 dataset. We randomly select three users and compare the distribution of their positive sample representations in the test set, which are displayed under the BoostGCN and LightGCN models at $K = 4$ respectively. As shown in Figure 5, two observations are found: 1) For LightGCN, the dispersed and sparse distribution of positive samples among these users suggests that the model's suppression-based aggregation method may struggle to capture their full preferences, potentially biasing node representations. 2) Conversely, BoostGCN effectively concentrates the positive samples of the same users into a compact space, demonstrating its ability to capture comprehensive user preferences. This capability underscores BoostGCN's superior performance in providing both comprehensive and personalized recommendations.

## 5.5 Analysis of Popularity Debiasing and Noise Resistance (RQ5)

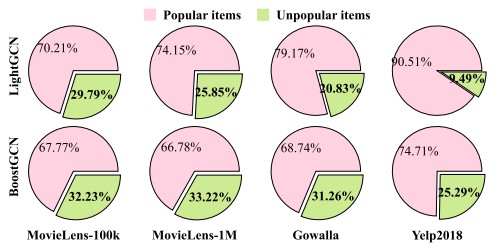

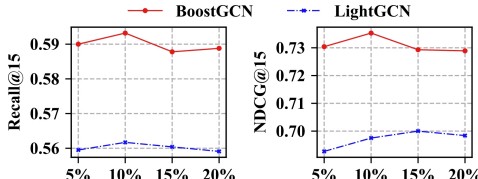

Figure 6: Proportion of popular and unpopular items in Top@15. BoostGCN is able to recommend around 2%-16% more unpopular items.

Figure 7: Performance comparison between BoostGCN ($\beta = e$) and LightGCN at different noise levels on ML-100k

**Popularity debiasing:** To show the popularity debiasing performance of BoostGCN, we show the results of Top@15 by the proportion of popular and unpopular items (the statistical tests across four datasets all yield p-value < 0.01), as shown in Figure 6. Specifically, we categorize the positive samples of each user in the test set based on interaction numbers, with the top 50% as popular items and the bottom 50% as unpopular items. Experiments show that BoostGCN is able to recommend about **2%-16%** more unpopular items, achieving a better performance of popularity debiasing. **Noise resistance:** To evaluate the noise resistance of BoostGCN, we randomly added 5%-20% spurious interactions to the dataset and compared it with LightGCN, as shown in Figure 7. We can see that BoostGCN outperforms LightGCN at different noise levels, demonstrating its superior noise resistance.

## 5.6 Limitations

BoostGCN is designed based on static graphs, which means it has significant room for improvement when dealing with real-time data. In Appendix A.7, we additionally examine BoostGCN's transfer potential and generalization scenarios across diverse domains. The amplification function we proposed performs best in the current exploration phase but is not necessarily optimal. Future research can further explore different amplification functions to achieve better performance, efficiency, and recommendation fairness. Notably, BoostGCN is designed to replace the base model LightGCN. Therefore, it can be combined with other advanced techniques to further improve performance.

## 6 Conclusion

In this paper, we propose a novel linear GCN model, namely BoostGCN. This model abandons the conventional approach of suppressing information propagation via the graph Laplacian norm, and instead explores and amplifies significant interactions within the interaction graph, thereby achieving a dual optimization of performance and efficiency. Also, we provide a new method for quantifying significant interactions. BoostGCN has relatively fixed parameters and stable performance, which is convenient for future improvements and applications. Notably, our visualization experiments ensure that BoostGCN is free from bias in recommendations.

## Acknowledgements

This work was supported by the National Natural Science Foundation of China (72271063, 71871069), and Guangdong Province Philosophy and Social Science Planning 2024 Annual General Project (GD24CGL45).

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

# A Appendices for BoostGCN

## A.1 Detailed Explanation for Related Concepts

**C.1**: '**Interaction significance**' is defined as the degree to which a user is likely to choose an item, which is influenced by various factors such as the item's quality, trust, price, etc [27, 26, 28]. For example, in movie recommendation, among horror movies, those with higher quality are more likely to be chosen by users. The quality of a movie can be partly obtained from modal information and partly inferred from interaction data. In this paper, we focus on mining such hidden information from interaction data.

**C.2**: '**Factors**' are defined as a broad term encompassing all factors that may influence a user's choice of an item, such as trust, quality, conformity, price and etc [27, 26, 28].

**C.3**: '**Key factors**' are defined as those factors that can be quantified given the available data content. For example, in this paper we focus only on user-item interaction data; hence, the factors that can be quantified through interactions constitute the key factors of this study.

**C.4**: '**The more interactions an item has, the higher a user is associated with the item.**' This finding is derived from a synthesis of relevant literature and real-world observations. For example, Yu et al. [26] have shown that items with more user interactions tend to induce conformity in users, thereby increasing the likelihood that these items will be chosen. In addition, when purchasing goods, items with a higher number of interactions can give users an impression of high quality and trustworthiness, which in turn encourages them to choose these items. Therefore, the ranking of the degree to which each item is chosen by users (referred to as "interaction significance") is consistent with the ranking of the number of interactions, i.e. $\mathcal{S}_{i \to u} \propto |\mathcal{N}_i|$. However, the increase in item quality, trustworthiness and other related factors is not linearly related to the increase in the degree of user choice. For this reason, we use a logarithmic function to quantify this hidden information in order to reduce errors.

**C.5**: '**The benefits of Amplification.**' From C.4, our amplification method compensates for the loss of information caused by the information suppression in the GCN basic framework. Notably, our method outperforms the most commonly used GCN frameworks in terms of recommendation fairness and noise resistance, as shown in Figures 6 and 7.

## A.2 The Selection and Advantages of Logarithmic Amplification Functions

**a) Reasons for amplifying first-order neighbors.** From the perspective of spectral graph convolution, the standard Laplacian-normalized aggregation is $\mathbf{e}_u^{(k+1)} = \sum_{i \in \mathcal{N}_u} \frac{1}{\sqrt{|\mathcal{N}_u|}\sqrt{|\mathcal{N}_i|}} \mathbf{e}_i^{(k)}$, where the significance of each item is $\frac{1}{\sqrt{|\mathcal{N}_u|}\sqrt{|\mathcal{N}_i|}}$. This aggregation will suppress those items with more significant interactions. However, we instead use $\frac{log_\beta(|\mathcal{N}_i|)+1}{\sqrt{|\mathcal{N}_i|}\sqrt{|\mathcal{N}_i|}}$, which can emphasize neighbors with more significant interactions, thereby accelerating the convergence of sparse interaction patterns and improving the quality of the representation.

**b) The selection criteria for the amplification function.** To balance the signal gain, robustness and learnability of significant interactions, we propose the following three criteria for selecting a node amplification function:

(i) it should increase monotonically with the number of interactions so that nodes with a high number of interactions contribute more information;

(ii) it should exhibit non-linear growth to prevent over-amplifying extremely interactive nodes and introducing noise;

(iii) it should be a closed-form, differentiable expression for efficient gradient-based optimization.

However, the linear amplification function violates (ii); the exponential amplification function violates (ii) and (iii). Consequently, the logarithmic amplification function is the optimal closed-form solution under these criteria.

**c) Advantages of the logarithmic amplification function.** The logarithmic amplifier in Eq.(8) exposes a single scalar $\beta$ that instantiates a tunable trade-off between magnitude amplification and

model sensitivity. A larger $\beta$ compresses inter-item differences $\varepsilon_{(i_p, i_q)}^{Amp}$, yielding a more uniform signal landscape, whereas a smaller $\beta$ stretches these gaps, heightening sensitivity to subtle interaction sparsity.

**Proof.**

Let interaction amplification be $\mathcal{S}_{i \to u}^{Amp} = log_\beta(|\mathcal{N}_i|) + \triangle^{log}$ with $\beta > 1$ and $\triangle^{log} > 0$. For any two neighbor items $i_p$ and $i_q$, define the difference $\varepsilon_{(i_p, i_q)}^{Amp} = log_\beta(|\mathcal{N}_{i_p}|) - log_\beta(|\mathcal{N}_{i_q}|)$. Taking the derivative w.r.t. $\beta$ gives $\frac{d\varepsilon^{Amp}}{d\beta} = \frac{1}{\beta}(ln(|\mathcal{N}_{i_p}|) - ln(|\mathcal{N}_{i_q}|))$. This derivative monotonically decreases with $\beta$, indicating that the greater $\beta$, the gentler the difference $\varepsilon^{Amp}$. The smaller $\beta$ is, the stronger the difference is magnified. $\qquad\square$

**d) Performance comparison with other amplification functions.** To further verify that the logarithmic amplification function is superior to other amplification functions, we compare the logarithmic amplification function with other amplification function variants: **linear** ($|\mathcal{N}_i|$), **square-root** ($\sqrt{|\mathcal{N}_i|}$) and **exponential** ($exp(|\mathcal{N}_i|)$) variants, as shown in Figure 8.

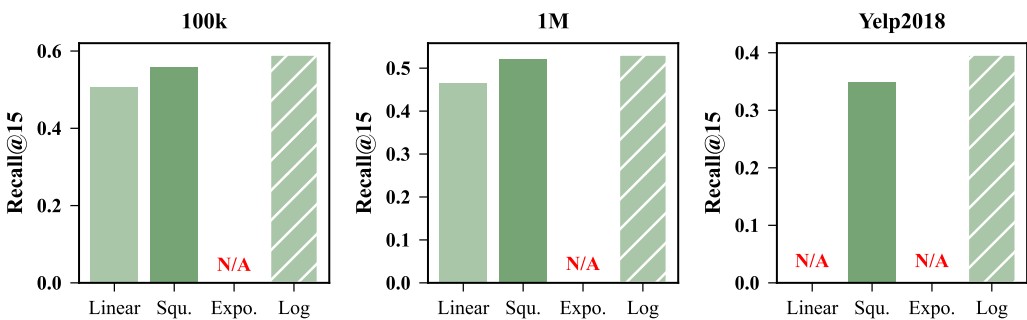

Figure 8: Performance comparison with other amplification functions.

From Figure 8, we can know that the linear and exponential amplification may cause gradient explosion and unstable training. Square-root amplification is relatively mild, but its performance is poor. The logarithmic amplification function provides the best balance among performance and stability.

## A.3 Advantages of BoostGCN's Aggregation over the Aggregation using GAT

In this section, we provide a systematic, theoretical comparison of BoostGCN and GAT from three perspectives: expressive power, sample complexity, and stability.

**a) Expressive Power**

Here we use the Vapnik-Chervonenkis (VC) dimension to measure the expressive power of a function class. The higher the VC dimension, the more complex the model becomes and the more likely it is to overfit on a limited number of samples. When VC dimension is zero, the model contains no learnable parameters and has the lowest complexity.

· GAT learns attention $\alpha_{ui} = softmax(LeakyReLu(\theta^T[W\mathbf{e}_u||W\mathbf{e}_i]))$, relying on the learnable parameter $\theta$ and the feature mapping $W$.

· The weight of BoostGCN is determined only by $w = log_\beta(|\mathcal{N}_v|)$ ($|\mathcal{N}_v|$ represents the node degree) and does not rely on additional learnable parameters. Therefore, the VC dimension is lower and it is less likely to overfit in scenarios with sparse and node-free features.

**Statement 1** (*Expressive Power under Feature Scarcity*): When $d_{feat} < log(n)$ ($d_{feat}$ represents the feature dimension of each node and $n$ represents the total number of nodes), GAT collapses to uniform weighting, whereas BoostGCN retains degree-based discrimination.

**Proof.**

From [40], the VC-dimension of a soft-attention network with $p$ learnable parameters is lower-bounded by $\Omega(plogn)$. GAT's parameters $p_{GAT} \geq d_{feat} \Rightarrow VC_{GAT} = \Omega(d_{feat}logn)$. When $d_{feat} < logn, VC_{GAT} \leq VC_{uniform}$, i.e., GAT collapses to uniform weighting. BoostGCN uses fixed weights $w = log_\beta(|\mathcal{N}_v|)$; no additional parameters $\Rightarrow VC_{BoostGCN} = 0$. Therefore, BoostGCN retains degree-based discrimination. □

**b) Sample Complexity**

· The attention parameter of GAT requires $\Omega(nlogn)$ interaction samples to converge to a stable weight with a high probability. BoostGCN only requires $O(n)$ samples to achieve the same generalization error because the weights are directly given by the degrees.

**Statement 2** (*Sample Complexity*): To achieve error $\leq \varepsilon$ with probability $\geq 1 - \delta$:

(1) GAT requires $m_{GAT} = \Omega(nlogn)$ interactions.

(2) BoostGCN requires $m_{Boost} = O(n)$ interactions.

**Proof.**

GAT: When the number of GAT's parameters is $p_{GAT} \geq d_{feat}$, standard VC bound [40] gives $m_{GAT} \geq \frac{1}{\varepsilon^2}(p_{GAT}logn + log\frac{1}{\delta}) = \Omega(nlogn)$.

BoostGCN: The weights of BoostGCN are directly given by degrees and are equivalent to zero parameters. The error is only derived from the empirical estimation of the degree. By Hoeffding's inequality, $m_{Boost} \geq \frac{1}{\varepsilon^2}log\frac{1}{\delta} = O(n)$. □

**c) Stability under Noise**

The sensitivity of GAT's attention weights to noise edges increases linearly. The logarithmic amplification of BoostGCN suppresses high-order noise in a sublinear manner, thus being more robust in sparse and noisy scenarios.

**Statement 3** (*Stability under Noise*): Let each edge be perturbed independently with probability $q$.

(1) GAT's attention changes satisfy: $\mathbb{E}||\alpha^{GAT} - \tilde{\alpha}^{GAT}||_1 \leq C_{GAT}qn$.

(2) BoostGCN's changes satisfy: $\mathbb{E}||w^{Boost} - \tilde{w}^{Boost}||_1 \leq C_{Boost}qlnn$.

Proof.

GAT: Attention weight change grows linearly with the expected number of noisy edges $qn$.

BoostGCN: The estimation error of the weight only depends on the degree. By Hoeffding's inequality, $\mathbb{E}|log_\beta(|\mathcal{N}_v|) - log_\beta(|\tilde{\mathcal{N}}_v|)| \leq \frac{lnn}{\sqrt{m}}$ (where $m$ refers to the total number of observed edges), yielding sub-logarithmic total error $\propto qlnn$.

**d) Overall Positioning**

· Sparse graphs, scarce features, or high noise: GAT's "learning" becomes a burden; BoostGCN offers parameter-free, sample-efficient, noise-robust weighting.

· Dense graphs with rich features: GAT remains complementary; BoostGCN can be freely combined with GAT.

## A.4 The Proofs of BoostGCN's Properties

**Proposition A.1** (*Smaller aggregation error*) *With BoostGCN's aggregation in Eq.(12), the error bound decays at the rate of $O(log_\beta(|\mathcal{N}_i|)+1)$ as the neighbor cardinality $|\mathcal{N}_i|$ increases, guaranteeing diminishing aggregation error even for highly interactive nodes.*

**Proof.**

Let $\mathcal{N}_u$ be the 1-hop neighbors of user $u$, $w_i = \frac{log_\beta(|\mathcal{N}_i|)+1}{|\mathcal{N}_u|}$ be the amplified weight, $\mathbf{e}_i^{tru}$ be the ground-truth embedding of item $i$, and $\mathbf{e}_i^{(k)}$ be the embedding after $k$ layers. Therefore, the aggregation error can be defined as $\varepsilon_u^{(k+1)} = ||\mathbf{e}_u^{(k+1)} - \mathbf{e}_u^{tru}||$, with $\mathbf{e}_u^{tru} = \sum_{i \in \mathcal{N}_u} w_i\mathbf{e}_i^{tru}$.

· First, we can get error propagation as follows: $\mathbf{e}_u^{(k+1)} = \sum_{i \in \mathcal{N}_u} w_i \mathbf{e}_i^{(k)} \Rightarrow \varepsilon_u^{(k+1)} = ||\sum_{i \in \mathcal{N}_u} w_i^2 (\mathbf{e}_i^{(k)} - \mathbf{e}_i^{tru})|| \leq \sum_{i \in \mathcal{N}_u} w_i \varepsilon_i^{(k)}$

· Next, we can assume that layer-wise error satisfies $\varepsilon_i^{(k)} \leq C \cdot \delta^{(k)}$ (consistent with LightGCN-style linear-GCN convergence analysis, $\delta < 1$ and $C$ is general constant).

· Then, because $w_i = \frac{log_\beta(|\mathcal{N}_i|)+1}{|\mathcal{N}_u|}$ and $log_\beta(|\mathcal{N}_i|) \leq log_\beta(max_j|\mathcal{N}_j|)$, we have $\sum_{i \in \mathcal{N}_u} w_i \leq \frac{log_\beta(max_j|\mathcal{N}_j|)+1}{|\mathcal{N}_u|} \cdot |\mathcal{N}_u| = log_\beta(max_j|\mathcal{N}_j|) + 1$.

· Finally, we combine the bounds $\varepsilon_u^{(k+1)} \leq \sum_{i \in \mathcal{N}_u} w_i \varepsilon_i^{(k)} \leq (log_\beta(max_j|\mathcal{N}_j|) + 1) \cdot C \cdot \delta^{(k)} = O(log_\beta(|\mathcal{N}_i|) + 1) \cdot \delta^{(k)}$.

Hence, the aggregation error bound decays at the rate of $O(log_\beta(|\mathcal{N}_i|)+1)$ as the neighbor cardinality $|\mathcal{N}_i|$ increases. $\qquad\square$

**Proposition A.2** *(Near-optimal provable linear convergence rate) By using the logarithmic amplifier in Eq.(8), BoostGCN can achieve a spectral-radius bound arbitrarily close to the infimum of the sub-linear family and yield a near-optimal provable linear convergence rate.*

Let node degree be $d$ and the information weight be $w(d)$, among the sub-linear family $F = \{w|w(d) = d^\alpha, 0 < \alpha < 1\}$, the logarithmic function $w(d) = log_\beta(d)$ exhibits the same asymptotic growth order as the limiting behavior of $F(\alpha \to 0)$, thereby achieving a spectral-radius bound arbitrarily close to the infimum of the family and yielding near-optimal provable linear convergence rate.

**Proof.**

· Consider the family of functions $F = \{w|w(d) = d^\alpha, 0 < \alpha < 1\}$.

· For $w_\alpha(d) = d^\alpha$, the corresponding aggregation matrix $W_\alpha = D^{-1} \cdot diag(d_i^\alpha) \cdot A$.

· Its infinite norm $||W_\alpha||_\infty = max_i \sum_j d_j^\alpha / d_i \leq d_{max}^\alpha$.

· For $w_\alpha(d) = d^\alpha$, the spectral radius satisfies $\rho(W_\alpha) \leq ||W_\alpha||_\infty \leq d_{max}^\alpha$ (Gershgorin's direct inference), which is monotonically increasing in $\alpha$.

· When $\alpha \to 0$, $d_{max}^\alpha \to 1$ is at its minimum. When $\alpha \to 1$, $d_{max}^\alpha \to d_{max}$ is at its maximum.

· The logarithmic weight $w(d) = log_\beta(d)$ at the limit of $\alpha \to 0$ satisfies $w(d) = lim_{\alpha \to 0+}(d^\alpha - 1)/\alpha \cdot 1/ln\beta \approx (d^\alpha - 1)/\alpha$ (first-order Taylor expansion, $\beta \to e$), hence it inherits the minimal growth rate of the family $F$ in the limit.

Among the sub-linear family $F$, the bound $\rho(W_\alpha)$ is minimized as $\alpha \to 0$. The logarithmic function exhibits the same growth order as this limit, thus achieving the tightest bound. $\qquad\square$

**Theorem A.3** *Under independent and identically distributed (i.i.d.) mini-batch sampling, bounded gradients and L-smoothness, BoostGCN's expected error bound is*

$$\mathbb{E}[||H^{(K)} - H^{tru}||_F^2] \leq (1 - 2\eta\lambda)^K \Delta_0 + \eta G^2/(2\lambda) \tag{19}$$

*And BoostGCN's high-probability bound is*

$$||H^{(K)} - H^{tru}||_F = O((log_\beta(max_j|\mathcal{N}_j|)+1)^K + \sqrt{\frac{log\frac{1}{\delta}}{K}}), \, with \, probability \, at \, least \, 1 - \delta. \tag{20}$$

*where $H^{(K)}$ is the current embedding matrix of all users and items after $K$ rounds update of random training and $H^{tru}$ is the ideal embedding when the loss function converges to the minimum value, where $H^{(K)}, H^{tru} \in \mathbf{R}^{(|\mathcal{U}|+|\mathcal{I}|) \times d}$; $\eta$ is the learning rate and $\lambda$ is the regularization term; $G$ is the boundary of gradients, where $G > 0$ and $||\nabla\mathcal{L}|| \leq G$ for any mini-batch; $\Delta_0 = \mathbb{E}[||H^{(0)} - H^{tru}||_F^2]$ and $\delta \in (0, 1)$ is confidence parameter.*

**Explicit assumptions:**

A1. **Independent and identically distributed (i.i.d.) mini-batch sampling**: The positive and negative samples in each mini-batch are independently and identically distributed from the true distribution;

A2. **Bounded gradients**: there exists $G > 0$ such that $||\nabla \mathcal{L}|| \leq G$ for any mini-batch;

A3. **L-smoothness**: the loss function is L-Lipschitz continuous in the embedding space.

**Notation**:

· The embedding matrix is $H \in \mathbf{R}^{(|\mathcal{U}|+|\mathcal{I}|) \times d}$.

· The regularization loss is $\mathcal{L}(H) = \mathcal{L}_{rec}(H) + \lambda ||H||_F^2$, where $\lambda > 0$ makes loss $\lambda$-strongly convex.

· Suppose A1-A3: i.i.d. sampling, gradient norm $\leq G$, L-Lipschitz continuous gradient.

· The learning rate $\eta$ is $\leq 1/\mathcal{L}$.

· The amplification matrix $W$ satisfies $\rho(W) \leq (log_\beta(max_j |\mathcal{N}_j|) + 1)$ (previously proved).

· $Z_k$ represents the martingale difference gradient noise generated by the $k$-th mini-batch.

· $M_K$ represents the cumulative random error matrix at the end of the $K$ iteration, which is the sum of all the mini-batch noises weighted by the amplification matrix $W$.

· $J$ stands for identity matrix.

**Proof.**

First, we prove the **BoostGCN's expected error bound**:

· Single-step update $H^{(k+1)} = H^{(k)} - \eta \nabla \mathcal{L}_{B_k}(H^{(k)})$, where $B_k$ is the mini-batch corresponding to the $k$-th iteration.

· Under assumptions A1–A3 and $\lambda$-strongly convexity, one step yields

$$
\begin{aligned}
&\mathbb{E}[||H^{(k+1)} - H^{tru}||_F^2 | H^k] \\
&= ||H^{(k)} - H^{tru}||_F^2 - 2\eta < H^{(k)} - H^{tru}, \nabla \mathcal{L}(H^{(k)}) > + \eta^2 \mathbb{E}[||\nabla \mathcal{L}_{B_k}(H^{(k)})||_F^2] \\
&\leq (1 - 2\eta\lambda)||H^{(k)} - H^{tru}||_F^2 + \eta^2 G^2
\end{aligned}
\tag{21}
$$

· Let $\Delta_k = \mathbb{E}[||H^{(k)} - H^{tru}||_F^2]$, then $\Delta_{k+1} \leq (1 - 2\eta\lambda)\Delta_k + \eta^2 G^2$.

· For $k = 0, 1, ..., K - 1$,

$$
\begin{aligned}
\Delta_K &\leq (1 - 2\eta\lambda)^K \Delta_0 + \eta^2 G^2 \sum_{t=0}^{K-1} (1 - 2\eta\lambda)^t \\
&= (1 - 2\eta\lambda)^K \Delta_0 + \eta^2 G^2 (1 - 2\eta\lambda)^K / (2\eta\lambda) \\
&\leq (1 - 2\eta\lambda)^K \Delta_0 + \eta^2 G^2 / (2\eta\lambda) \\
&= (1 - 2\eta\lambda)^K \Delta_0 + \eta G^2 / (2\lambda)
\end{aligned}
\tag{22}
$$

Therefore, the expected error bound of BoostGCN is $\mathbb{E}[||H^{(K)} - H^{tru}||_F^2] \leq (1 - 2\eta\lambda)^K \Delta_0 + \eta G^2 / (2\lambda)$.

Next, we prove the **BoostGCN's high-probability bound**:

· Martingale Difference Sequence:

$$
Z_k = \nabla \mathcal{L}_{B_k}(H^{(k)}) - \nabla \mathcal{L}(H^{(k)}), \ \ \mathbb{E}[Z_k] = 0, \ \ |Z_k|_F \leq G
\tag{23}
$$

· Cumulative Error:

$$
M_K = \sum_{k=0}^{K-1} (J - \eta W)^{K-k-1} Z_k
\tag{24}
$$

· Norm Bound:

$$||(J - \eta W)^{K-k-1} Z_k||_F \leq (\rho(W))^{K-k-1} G \leq (log_\beta(max_j|\mathcal{N}_j|) + 1)^{K-k-1} G \quad (25)$$

· Vector Azuma–Hoeffding Inequality:

$$\mathbb{P}(||M_K||_F \geq t) \leq 2exp(-\frac{t^2}{2KG^2}) \quad (26)$$

· Total Error Decomposition:

$$||H^{(K)} - H^{tru}||_F \leq (log_\beta(max_j|\mathcal{N}_i|) + 1)^K ||H^{(0)} - H^{tru}||_F + ||M_K||_F \quad (27)$$

· Final High-Probability Bound: Let $t = G\sqrt{2Klog\frac{2}{\delta}}$, then

$$\mathbb{P}(||H^{(K)} - H^{tru}||_F \leq (log_\beta(max_j|\mathcal{N}_i|) + 1)^K ||H^{(0)} - H^{tru}||_F + G\sqrt{2Klog\frac{2}{\delta}}) \geq 1 - \delta \quad (28)$$

Therefore, $||H^{(K)} - H^{tru}||_F = O((log_\beta(max_j|\mathcal{N}_i|) + 1)^K + \sqrt{\frac{log\frac{1}{\delta}}{K}}), w.p. \geq 1 - \delta.$ □

## A.5 Baselines

We compare some representative models, ranging from traditional matrix factorization models to state-of-the-art GCN-based models.

- **MF-BPR** [33] is a matrix factorization model optimized by a pairwise ranking loss in a Bayesian way.

- **MMGCN** [29] is a classic multimodal recommendation model. In the experiment, we only employ ID embedding as its input, denoted by MMGCN$^{id}$.

- **NGCF** [11] leverages collaborative signals from high-order connectivity of the user-item graph via feature transformation and nonlinear activation functions.

- **LightGCN** [12] is a linear GCN framework that uses only linear aggregation without feature transformation and nonlinear activation functions between neighbors. As it is the most basic GCN model, it serves as the main baseline for our comparison.

- **UltraGCN** [13] with additional constraint loss further simplifies LightGCN by removing the stacking of many graph convolution layers in GCN.

- **IMP-GCN** [34] categorizes users into unique subgroups aligned with their particular interests, conducting advanced graph convolution operations exclusively within these subgroups.

- **NSE-GCN** [35] employs a neighborhood structure embedding technique that relies on first-order adjacency information to generate structural embeddings.

- **LayerGCN** [25] is state-of-the-art GCN model with the DegreeDrop mechanism, which refines layer representations during information propagation and node updating.

- **LTGNN** [36] performs linear-time graph convolution, scaling GNN-based recommendation to the efficiency of classic MF while preserving high-order connectivity modeling.

- **TransGNN** [37] alternates Transformer and GNN layers to globally enlarge the receptive field while disentangling edge-based aggregation.

- **GAT-LightGCN** combines LightGCN with Graph Attention Network.

In order to make a fair comparison, we carefully tune the hyper-parameters of each model based on their respective published papers and hyper-parameter studies. All baselines compared in this paper can be obtained directly from the corresponding literature.

## A.6 Datasets

To comprehensively demonstrate the effectiveness of BoostGCN, we evaluate our model on four distinct datasets, including MovieLens-100k (denoted by 100k) [29], MovieLens-1M (denoted by 1M) [29], Gowalla (denoted by Gowa.) [30] and Yelp2018 (denoted by Yelp) [11], as detailed in Table 2. The datasets utilized in this paper are all publicly available and can be directly downloaded from their respective sources.

The descriptions of these datasets are given below:

- **MovieLens Dataset** [29]: This dataset has been widely used for the recommendation evaluation, and it contains a series of subsets such as MovieLens-100k and MovieLens-1M.

- **Gowalla Dataset** [30]: This dataset is a check-in dataset obtained from Gowalla, in which users share their locations through checking-in.

- **Yelp2018 Dataset** [11]: This dataset is adopted from the 2018 edition of the Yelp challenge. The local businesses like restaurants and bars are considered as items.

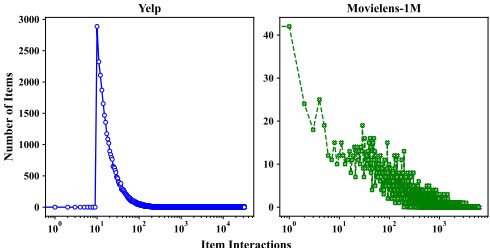
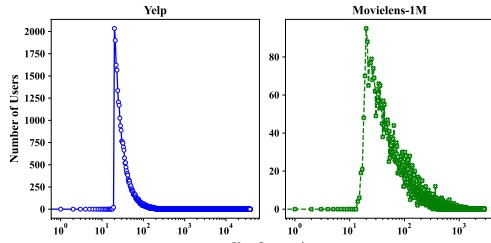

Figure 9: Long-tail items on Yelp and ML-1M datasets.

Figure 10: Long-tail users on Yelp and ML-1M datasets.

The reason why we chose these four datasets is that they vary greatly in size and sparsity, which is more conducive to demonstrating that our method is effective in different scenarios. For example, it can be seen from Figure 9 and Figure 10 that the long-tail problem of Yelp is much more severe than that of ML-1M. However, as shown in Table 3 and Table 4 of our paper, our performance on Yelp has improved the most, indicating that our performance can still be excellent when faced with the long-tail problem.

## A.7 Discussion on the Potential Applications of BoostGCN

### A.7.1 The Effect of Logarithmic Amplification Function on Non-linear GCN

Since BoostGCN's log-amplification is a linear re-weighting scheme, it can be grafted onto any non-linear GCN without structural changes-simply replace the original Laplacian weights with the BoostGCN weights, while keeping all non-linear activations, feature transformations and residual connections intact. To validate this, we conduct new experiments, as shown in Figure 11. We take NGCF (with Leaky-ReLU and feature transformation) and substitute its Laplacian weights with BoostGCN weights. The value of the amplification function $\beta$ is searched from $e$ to $4e$, and other settings remain consistent with NGCF. We can find that adding our amplification function to the four datasets will all improve the model performance. Thus, BoostGCN is readily compatible with non-linear GCNs and yields consistent gains.

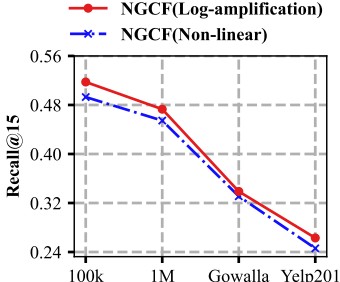

Figure 11: The performance comparison between NGCF and NGCF with our log-amplification on four datasets.

### A.7.2 Discussion on Dynamic Graph Scenarios

Remaining purely static in highly dynamic scenarios (real-time recommender systems, traffic forecasting) risks: **1) Concept drift**: user interests or item popularity can shift abruptly, causing performance drops. **2) Stale recommendations**: outdated interactions are over-amplified, degrading user experience. Therefore, we can extend BoostGCN to dynamic graphs via two complementary mechanisms:

• **Time-aware amplification function**: we plan to incorporate the interaction timestamp $t$ into saliency:

$$S_{(i \to u,t)}^{Amp} = log_\beta(|\mathcal{N}_i(t)|) \cdot exp(-\lambda(t - t_{last})) + \Delta^{log} \tag{29}$$

where $|\mathcal{N}_i(t)|$ counts interactions up to time $t$, $\lambda$ controls temporal decay, and $\Delta^{log}$ retains the offset. This lets recent interactions dominate while older ones gradually fade.

• **Dynamic adjacency caching & incremental updates**: upon each new interaction $(u, i, t)$, we perform a rank-1 update on the local adjacency matrix and reuse the incremental-propagation framework of LightGCN [41] to avoid full-graph re-computation, maintaining efficiency on par with the static version.

### A.7.3 BoostGCN: From Recommendation to General Graph Representation

• **Generalizing the assumption**: we show that the amplification weight $w_i \propto log(|\mathcal{N}_i|)$ is mathematically a smooth re-scaling of node degree. Degree is a universal structural cue independent of the "item-selection" semantics. Hence, in other domains one can simply replace $|\mathcal{N}_i|$ with any relevant structural count (e.g., number of friends in social networks, entity frequency in knowledge graphs) without changing the framework.

• **Robustness when the assumption is relaxed**: Figure 7 reports a ablation where we randomly increase noise interactions by 5% to 20% in the 100k dataset, breaking the original correspondence between high interaction and high saliency. The performance of BoostGCN is still superior to that of LightGCN, which indicates that our proposed method is resilient to violations of the behavioral prior.

• **Future extension**: we now state in the conclusion that if domain priors are unavailable, one can substitute $log(|\mathcal{N}_i|)$ with parameter-free alternatives such as PageRank [42] scores or self-supervised edge-importance estimates without altering the BoostGCN pipeline or complexity.

### A.7.4 Future Work

Although our work focuses on recommendation, the core idea of BoostGCN—amplifying salient first-order interactions via a logarithmic weighting function—is a general graph-aggregation principle. Consequently, the method is directly applicable to any task that requires node representation learning with edge-importance weighting, such as:

• influence maximization in social networks (up-weighting trust edges) [43];

• molecular property prediction (emphasizing bonds directly connected to functional groups) [44, 45];

• traffic flow forecasting (amplifying interactions with frequently used neighboring road segments) [46, 47].

