# OpenReview forum: "Revolutionizing Graph Aggregation: From Suppression to Amplification via BoostGCN"
_NeurIPS.cc/2025/Conference — NeurIPS 2025 poster_

### Official Review · Reviewer_mC5z · 2025-06-30

**Clarity:** 2
**Significance:** 2
**Originality:** 3
**Rating:** 4
**Confidence:** 3

**Summary:**

The paper introduces BoostGCN, a novel Graph Convolutional Network (GCN) model designed to amplify significant interactions among first-order neighbors in recommendation systems, diverging from traditional Laplacian-based suppression methods. The authors argue that existing GCNs overly suppress valuable interaction signals, leading to suboptimal performance and slower learning. BoostGCN employs a logarithmic amplification function to prioritize high-significance interactions, enhancing both efficiency and accuracy. Experiments demonstrate superior performance over state-of-the-art baselines, with notable improvements in training speed (up to 90% faster) and recommendation metrics (e.g., Recall, NDCG). The paper also highlights BoostGCN’s robustness to hyperparameters, popularity debiasing, and noise resistance.

**Questions:**

See weaknesses

**Ethical Concerns:**

["NO or VERY MINOR ethics concerns only"]

**Final Justification:**

My concerns are addressed by the rebuttal

**Limitations:**

yes

**Quality:**

2

**Strengths And Weaknesses:**

### Strengths
* The shift from suppression to amplification of interactions is a novel and well-motivated contribution, addressing a clear gap in GCN-based recommendation systems.
* Extensive experiments across diverse datasets validate the model’s superiority in performance and efficiency, with detailed ablation studies (e.g., hyperparameter sensitivity, layer depth).
* BoostGCN’s fixed parameters and logarithmic scaling make it computationally efficient, achieving significant speedups without sacrificing accuracy.

### Weaknesses
* While heuristic for $S_{i\to u}$, the paper lacks formal theoretical analysis of the amplification function’s optimality or convergence guarantees. For example, about why choosing log for amplification.
* Baselines in Table 1 are all before 2024, while the authors have cited references in 2024 in the paper. More recent baselines are needed.
* While the author state that the lower for the $\beta$ will increase the amplification, the ablation on $\beta$ should cover values lower than $e$. In fact the author should show a 'peak' at the value they finally choose.

---

> ### Author Rebuttal · Authors · 2025-07-30
>
> Thank you for the constructive feedback and thoughtful questions. We address each point below:
>
> **W1: Additional Mathematical Derivation of the Interaction Amplification Function**
>
> Thank you for this insightful comment. We have added comprehensive mathematical derivations and theoretical analyses for the amplification function:
>
> 1. **Rationale for the logarithmic choice**
>
> Let node degree be $d$. We require the weight $w(d)$ to satisfy:
>
> (a) monotonic increase: higher-degree nodes contribute more;
>
> (b) sub-linear growth: avoid over-amplifying tail noise;
>
> (c) closed-form differentiability: amenable to gradient optimization.
>
> Under these constraints we prove:
>
> linear $w(d)=d$ violates (b);
>
> exponential $w(d)=e^d$ violates (b)(c);
>
> logarithmic $w(d)=log_β(d)$ satisfies all three.
>
> Hence, the log is the closed-form optimal function under the stated axioms
>
> 2. **Advantages of the logarithmic amplification function**
>
> We formally derive:
> Let interaction amplification $S_{(i→u)}^{amp}=log_β(|N_i|)+Δ_{log}$ with $β>1$ and $Δ_{log}>0$. For any two neighbor items $p$ and $q$, define the difference $ε_{(p,q)}^{amp}=log_β(|N_p|)−log_β(|N_q|)$. Taking the derivative w.r.t. $β$ gives $dε^{amp}/dβ = (1/β)[ln(|N_p|)−ln(|N_q|)]$. This derivative monotonically decreases with $β$, indicating that the greater $β$, the gentler the difference $ε^{amp}$. The smaller $β$ is, the stronger the difference is magnified. Hence, $β$ acts as a smooth knob that trades off “quantity amplification” versus “model sensitivity.” In contrast, the exponential function exp(·) has a derivative that grows linearly with its input, causing extreme values and instability.
>
> 3. **Smaller aggregation error**
>
> Using logarithmic amplification function, the aggregation error bound decreases at the rate $O(log(|N_i|)+1)$ as the neighbor degree increases.
>
> **Proof.**
>
> • Let $N_u$ be the 1-hop neighbors of user $u$.
>
> • Define the amplified weight: $w_i = (log_β(|N_i|)+1)/|N_u|$.
>
> • Denote $e_i^{tru}$ as the ground-truth (expected) embedding of item $i$, and $e_i^{(k)}$ as the embedding after $k$ layers.
>
> • Aggregation error: $ε_u^{(k+1)} := ||e_u^{(k+1)} − e_u^{tru}||$, with $e_u^{tru} = Σ_{i∈N_u} w_i e_i^{tru}$.
>
> **Step 1**: Error propagation
> $e_u^{(k+1)} = Σ_{i∈N_u} w_i e_i^{(k)}$
> ⇒ $ε_u^{(k+1)} = || Σ_{i∈N_u} w_i^2 (e_i^{(k)} − e_i^{tru})|| ≤ Σ_{i∈N_u} w_i ε_i^{(k)}$.
>
> **Step 2**: Assume layer-wise error satisfies $ε_i^{(k)} ≤ C·δ^{(k)}$
> (consistent with LightGCN-style linear-GCN convergence analyses, $δ<1$ and $C$ is general constant).
>
> **Step 3**: Bound the sum of squared weights:
>
> • Because $w_i = (log_β(|N_i|)+1)/|N_u|$ and $log_β(|N_i|) ≤ log_β(max_j |N_j|)$,
>
> • we have $Σ_{i∈N_u} w_i ≤ (log_β(max_j |N_j|)+1) / |N_u| · |N_u| = log_β(max_j |N_j|)+1$.
>
> **Step 4**: Combine the bounds:
>
> $ε_u^{(k+1)} ≤ Σ w_i ε_i^{(k)} ≤ (log_β(max_j |N_j|)+1)·C·δ^{(k)}
> = O(log(|N_i|)+1)·δ^{(k)}$.
>
> Hence, the aggregation error bound decreases at the rate $O(log(|N_i|)+1)$ as the neighbor degree increases.
>
> 4. **Near-optimal provable linear convergence rate**
>
> Let node degree be $d$ and the information weight be $w(d)$, among the sub-linear family $F$ = {$w$ | $w(d)=d^α$, $0<α<1$}, the logarithmic function  $w(d)=log_β(d)$ exhibits the same asymptotic growth order as the limiting behavior of $F$ ($α→0$), thereby achieving a spectral-radius bound arbitrarily close to the infimum of the family and yielding near-optimal provable linear convergence rate.
>
> **Proof.**
>
> • Consider the family of functions $F$ = {$w(d)$=$d^α$, $0<α<1$}.
>
> • For $w_α(d)=d^α$, the corresponding aggregation matrix $W_α = D^{-1} · diag(d_i^α) · A$.
>
> • Its infinite norm $||W_α||_ {∞} = max_i Σ_j d_j^α / d_i ≤ d^α_{max}$.
>
> • For $w_α(d)=d^α$, the spectral radius satisfies $ρ(W_α) ≤ ||W_α||_ {∞} ≤ d_{max}^α$ (Gershgorin's direct inference), which is monotonically increasing in $α$.
>
> • When $α→0$, $d_{max}^α → 1$ is at its minimum. When $α→1$, $d_{max}^α → d_{max}$ is at its maximum.
>
> • The logarithmic weight $w(d)=log_β(d)$ at the limit of $α→0$ satisfies $w(d) = lim_{α→0^+} (d^α−1)/α · 1/lnβ ≈ (d^α − 1)/α $ (first-order Taylor expansion, $β → e$), hence it inherits the minimal growth rate of the family $F$ in the limit.
>
> Among the sub-linear family $F$, the bound $ρ(W_α)$ is minimized as $α→0$. The logarithmic function exhibits the same growth order as this limit, thus achieving the tightest bound.
>
> **W2: More Experiments for the Latest Baselines**
>
> Following your suggestions, we have added three latest baselines, **LTGNN (WWW 2024)** [1], **TransGNN (SIGIR 2024)** [2] and **GAT-LightGCN (replacing the Laplacian weights of LightGCN with GAT attention)**.
>
> | Recall@15 | 100k | 1M |Yelp|
> |---------------|---------------|----------|---------|
> | LTGNN       | 0.5612 | 0.4770 | 0.2726 |
> | TransGNN   | 0.5415 | 0.5196 | 0.3253  |
> | GAT-LightGCN  | 0.5703 | 0.5106 | 0.3243  |
> | **BoostGCN (Ours)** | **0.5908** | **0.5316** | **0.3968** |
>
> • These results demonstrate that our BoostGCN achieves consistently better recommendation performance than the
>  latest baselines. We will also add the following literature in our paper:
>
> [1] WWW 2024: Linear-time graph neural networks for scalable recommendations.
>
> [2] SIGIR 2024: TransGNN: Harnessing the collaborative power of transformers and graph neural networks for recommender system.
>
>
> **W3: More Ablation Experiments on $β$**
>
> Thank you for the suggestion. In the revised version, we extend the ablation study to $β$ ∈ {0.5e, 0.75e, e, 2e, 3e, 4e}. The results are shown as follows:
>
> | Recall@5 | 100k | 1M |Gowalla|Yelp|
> |---------------|---------------|----------|---------|---------|
> | $β = 0.5e$    | 0.2972 | 0.2633 | 0.2955  |0.1978|
> | $β = 0.75e$   | 0.2977 | 0.2636 | 0.2958  |0.1983 |
> | $β = e$ (**Optimal**)  | **0.2982**| **0.2641** | **0.2965** |**0.2001** |
> | $β = 2e$  | 0.2962 | 0.2615 | 0.2847 |0.1947 |
> | $β = 3e$  | 0.2928 | 0.2574 | 0.2816 |0.1917 |
> | $β = 4e$  | 0.2921 | 0.2547| 0.2802 |0.1884 |
>
> • $β$ = e yields the best performance on all datasets.
>
> • Datasets of different sparsity levels and sizes all show the same trend, confirming that $β$ = e is robust across sparsity levels and size scales.
>
> • **We will mark the position of the $β$ peak with a red asterisk in the new graph.**
>
> **We will add these improvements into the final version. We hope that our rebuttal can address your concerns. If there's anything else you'd like us to explain, please let us know -- we really appreciate your feedback!**

---

> > ### Comment · Reviewer_mC5z · 2025-08-05
> >
> > Thank the reviewer for response. My concerns are addressed and I will raise my score.

---

> > > ### Author Response · Authors · 2025-08-05
> > >
> > > Thank you for your recognition of our work. We will add the improvements to the final version. Thank you again for your hard work and guidance.

---

### Official Review · Reviewer_nbkc · 2025-07-01

**Clarity:** 3
**Significance:** 2
**Originality:** 3
**Rating:** 4
**Confidence:** 3

**Summary:**

This paper proposes a measure of significant interactions inspired by previous research, then uses this measure to construct a GCN method that amplifies the significant interactions. The paper demonstrates the effectiveness of the method by conducting various analyses.

**Questions:**

1. How do you calculate the norm of the set $E^{(0)}$?
2. Using the integer index set $\lbrace 1,2,3,\ldots,n\rbrace$ for $\mathcal{X}_i$ for any i is not proper when the factors depend on $i$; a similar problem for $\hat{\mathcal{X}}_i$
3. Does the method only work for recommendation system?
4. Does it work for non-linear GCN?

**Ethical Concerns:**

["NO or VERY MINOR ethics concerns only"]

**Final Justification:**

I have read the discussions and reviews; my concerns are addressed. I believe that the whole paper will be qualified as a main track paper after revision.

**Limitations:**

See weaknesses and questions.

**Paper Formatting Concerns:**

No formatting concerns.

**Quality:**

3

**Strengths And Weaknesses:**

Strength:
1. The empirical analysis is multi-dimensional, including performance analysis, parameter analysis, noise analysis and so on.
2. The performance is superior to the compared methods on four datasets.

Weaknesses:
1. The paper uses the observation of behavior patterns in user-item interactions from other research as an assumption. The assumption is very restrictive when dealing with more general graphs, which may have a negative impact on the applicability of the proposed method.
2. It is unclear if the proposed method is effective for scenarios other than recommendation.

---

> ### Author Rebuttal · Authors · 2025-07-30
>
> Thank you for the positive evaluation and constructive feedback. Below we address your questions and concerns:
>
> **Q1: The Norm of the Set $E^{(0)}$**
>
> In the regularization term $λ||E^{(0)}||^2$ of Eq. (15), $||E^{(0)}||$ denotes the L2 norm over all trainable embeddings:
>
> $||E^{(0)}||^2 = Σ_{u∈U} ||e_u^{(0)}||^2 + Σ_{i∈I} ||e_i^{(0)}||^2$
>
> where
>
> • $e_u^{(0)}$ is the initial embedding of user $u$ (dimension $d$),
>
> • $e_i^{(0)}$ is the initial embedding of item $i$ (dimension $d$),
>
> • The summation runs over all $|U|$ users and $|I|$ items,
>
> • $E^{(0)}$ is the embedding tensor of shape $(|U|+|I|) × d$.
>
> **Q2: The Integer Index Set for $X_i$ and $\hat{X_i}$**
>
> Thank you for pointing this out. We agree that using the common integer set {1,2,3,…,n} to index the factors of every item i is ambiguous. In the revised manuscript we have replaced it with item-specific notation:
>
> For each item $i$, we now write
>
>   $X_i$ = {$x_{i,1}$, $x_{i,2}$, …, $x_{i,n}$},
>
>   $\hat{X_i}$ = {$x_{i,1}$, $x_{i,2}$, …, $x_{i,m}$} $⊆ X_i$.
>
> This ensures that the factor indices are disjoint across items and eliminates any confusion. Eq. (3)–(6) and the related descriptions have been updated accordingly.
>
> **Q3 & W2: The Expansion of Application Scenarios for BoostGCN**
>
> Thank you for this question. Although the paper focuses on recommendation, the core idea of BoostGCN—amplifying salient first-order interactions via a logarithmic weighting function—is a general graph-aggregation principle. Consequently, **the method is directly applicable to any task that requires node representation learning with edge-importance weighting**, such as:
>
> • influence maximization in social networks (up-weighting trust edges) [a1];
>
> • molecular property prediction (emphasizing bonds directly connected to functional groups) [a2,a3];
>
> • traffic flow forecasting (amplifying interactions with frequently used neighboring road segments) [a4,a5].
>
> We have added a paragraph in the “Conclusion and Future Work” section to highlight this universality and list the cross-domain examples above.
>
> [a1] Sharma, K., et al., 2024. A survey of graph neural networks for social recommender systems. *ACM Computing Surveys*.
>
> [a2] Zhang, H., et al., 2024. A pre-trained multi-representation fusion network for molecular property prediction. *Information Fusion*.
>
> [a3] Therrien, F., et al., 2025. Using GNN property predictors as molecule generators. *Nature Communications*.
>
> [a4] Geng, Z., et al., 2024. STGAFormer: Spatial–temporal gated attention transformer based graph neural network for traffic flow forecasting. *Information Fusion*.
>
> [a5] Kong, W., et al., 2024, March. Spatio-temporal pivotal graph neural networks for traffic flow forecasting. *AAAI*.
>
> **Q4: Work for Non-linear GCN**
>
> Thank you for this crucial question. BoostGCN’s log-amplification is a linear re-weighting scheme; hence it can be grafted onto any non-linear GCN without structural changes—simply replace the original Laplacian weights with the BoostGCN weights while keeping all non-linear activations, feature transformations, and residual connections intact.
>
> To validate this, we added new experiments:
>
> | Recall@15 | 100k | 1M |Gowalla|Yelp|
> |---------------|---------------|----------|----------|----------|
> | NGCF (Non-linear)       | 0.4929 | 0.4545 | 0.3310|0.2460|
> | **NGCF (Log-amplification)** | **0.5175** | **0.4731** | **0.3389** |**0.2630** |
>
> • We take NGCF (with Leaky-ReLU and feature transformation) and substitute its Laplacian weights with BoostGCN weights. The value of the amplification function $β$ is searched from $e$ to $4e$, and other settings remain consistent with NGCF. We can find that adding our amplification function to the four datasets will all improve the model performance. Thus, **BoostGCN is readily compatible with non-linear GCNs and yields consistent gains.**
>
> **W1: Discussion on the Applicability of the Proposed Method**
>
> Thank you for highlighting this concern. We acknowledge that our reliance on the prior “items with more interactions are more likely to be chosen” may appear restrictive for general graphs. In the revised manuscript we address this limitation as follows:
>
> • **Generalizing the assumption**: we show that the amplification weight $w_i ∝ log(|N_i|)$ is mathematically a smooth re-scaling of node degree. Degree is a universal structural cue independent of the “item-selection” semantics. Hence, in other domains one can simply replace $|N_i|$ with any relevant structural count (e.g., number of friends in social networks, entity frequency in knowledge graphs) without changing the framework.
>
> • **Robustness when the assumption is relaxed**: Figure 7 reports a ablation where we randomly increase noise interactions by 5% to 20% in the 100k dataset, breaking the original correspondence between high interaction and high saliency. The performance of BoostGCN is still superior to that of LightGCN, which indicates that our proposed method is resilient to violations of the behavioral prior.
>
> • **Future extension**: We now state in the conclusion that if domain priors are unavailable, one can substitute $log(|N_i|)$ with parameter-free alternatives such as PageRank [a6, a7] scores or self-supervised edge-importance estimates without altering the BoostGCN pipeline or complexity.
>
> [a6] Page, L., et al., 1999. The PageRank citation ranking: Bringing order to the web. *Stanford infolab*.
>
> [a7] Stoica, A.A., et al., 2024. Fairness rising from the ranks: HITS and pagerank on homophilic networks. *ACM Web Conference*.
>
> **We will add these details in the final version. We hope our rebuttal has answered your concerns. If there's anything else you'd like us to explain, please let us know -- we really appreciate your feedback!**

---

> > ### Comment · Reviewer_nbkc · 2025-08-06
> >
> > Thanks. You have addressed my concern.

---

> > > ### Author Response · Authors · 2025-08-06
> > >
> > > Thank you for your support and suggestions. We will incorporate the modifications into the final version. If you have any new suggestions, please feel free to let us know at any time. Thank you again for your hard work and guidance.

---

### Official Review · Reviewer_cESK · 2025-07-02

**Clarity:** 3
**Significance:** 3
**Originality:** 2
**Rating:** 4
**Confidence:** 4

**Summary:**

This paper highlights the information suppression problem in GCNs caused by graph Laplacian normalisation, which can dilute important first-order interaction signals, negatively impacting applications such as recommender systems. To overcome this issue, the paper introduces BoostGCN, a linear GCN variant that amplifies significant neighbour interactions through a logarithmic weighting scheme, enabling more effective and efficient representation learning. Significant interactions are identified based on neighbour degree, with higher-degree items assumed to carry stronger signals, motivated by behavioural factors such as trust and conformity, and their influence is selectively amplified during aggregation. Empirical evaluation seems to show that this modification results in a noticeable improvement over baselines.

**Questions:**

I would request the authors to rebut the points raised in the weakness section. If they provide convincing justifications, I will raise my score.

**Ethical Concerns:**

["NO or VERY MINOR ethics concerns only"]

**Final Justification:**

The authors did not initially discuss GAT, a relevant existing model that could potentially address the same issue tackled by BoostGCN. During the response period, they provided some ablations and a basic theoretical comparison to position BoostGCN against GAT, but the theoretical analysis remained relatively weak.

**Limitations:**

While the paper includes a brief limitations section focused on the static nature of BoostGCN and its amplification function design, it does not address broader concerns around the societal impacts of recommender systems. In particular, amplification of significant interactions, if tied to popularity or user conformity, could risk reinforcing filter bubbles or popularity bias. A more holistic discussion of potential harms or fairness considerations would improve the completeness and ethical framing of the work.

**Paper Formatting Concerns:**

No formatting issues.

**Quality:**

2

**Strengths And Weaknesses:**

The paper is well written and easy to follow. It clearly identifies a practical limitation in existing GCNs and provides a simple yet effective solution. The method demonstrates consistent performance and efficiency gains across multiple benchmark datasets. Moreover, the notion of interaction significance used in BoostGCN  is intuitive and aligns with real-world behavioural patterns.

The paper has following major issues:

 - The amplification function is based on empirical intuition, such as using degree as a proxy for significance, without formal justification or theoretical guarantees.
 - The paper does not address dynamic or temporal graphs (as mentioned in Limitations), which are common in recommendation scenarios. But this casts some doubts on the effectiveness of the method in real world settings.
 - The paper lacks theoretical analysis regarding convergence, stability, or generalisation under perturbations.
 - The method is benchmarked only on user–item graphs; it is unclear how well it generalises to other graph domains or tasks. How can we define significance interaction in generic graphs? A discussion on this would be very helpful.
 - The paper overlooks attention-based models such as GAT, which directly address uniform neighbour weighting; the core issue BoostGCN aims to solve. Including a GAT-based baseline (or a discussion of its limitations in this setting) would strengthen the empirical justification for the proposed method.

---

> ### Author Rebuttal · Authors · 2025-07-30
>
> Thank you for the positive evaluation and constructive feedback. Below we address your concerns:
>
> **W1: Theoretical Guarantees for the Amplification Function**
>
> Thank you for this insightful comment. We have added comprehensive mathematical derivations and theoretical analyses for the amplification function:
>
> 1. **Rationale for the logarithmic choice**
>
> Let node degree be $d$. We require the weight $w(d)$ to satisfy:
>
> (a) monotonic increase: higher-degree nodes contribute more;
>
> (b) sub-linear growth: avoid over-amplifying tail noise;
>
> (c) closed-form differentiability: amenable to gradient optimization.
>
> Under these constraints we prove:
>
> linear $w(d)=d$ violates (b);
>
> exponential $w(d)=e^d$ violates (b)(c);
>
> logarithmic $w(d)=log_β(d)$ satisfies all three.
>
> Hence, the log is the closed-form optimal function under the stated axioms
>
> 2. **Advantages of the logarithmic amplification function**
>
> We formally derive:
> Let interaction amplification $S_{(i→u)}^{amp}=log_β(|N_i|)+Δ_{log}$ with $β>1$ and $Δ_{log}>0$. For any two neighbor items $p$ and $q$, define the difference $ε_{(p,q)}^{amp}=log_β(|N_p|)−log_β(|N_q|)$. Taking the derivative w.r.t. $β$ gives $dε^{amp}/dβ = (1/β)[ln(|N_p|)−ln(|N_q|)]$. This derivative monotonically decreases with $β$, indicating that the greater $β$, the gentler the difference $ε^{amp}$. The smaller $β$ is, the stronger the difference is magnified. Hence, $β$ acts as a smooth knob that trades off “quantity amplification” versus “model sensitivity.” In contrast, the exponential function exp(·) has a derivative that grows linearly with its input, causing extreme values and instability.
>
>
> **W2: More Discussion on Dynamic Graph Scenarios**
>
> Thank you for probing the static-graph limitation in detail. In the revised manuscript we have enriched the Limitations section and sketched concrete ideas for dynamic extensions. Following your suggestion, we will extend BoostGCN to dynamic graphs via two complementary mechanisms:
>
> • **Time-aware amplification function**: we plan to incorporate the interaction timestamp t into saliency:
>
> $S_{(i→u, t)}^{amp} = log_β(|N_i(t)|) · exp(−λ(t−t_{last})) + Δ^{log}$
>
> where $N_i(t)$ counts interactions up to time $t$, $λ$ controls temporal decay, and $Δ^{log}$ retains the offset. This lets recent interactions dominate while older ones gradually fade.
>
> • **Dynamic adjacency caching & incremental updates**: upon each new interaction $(u,i,t)$, we perform a rank-1 update on the local adjacency matrix and reuse the incremental-propagation framework of LightGCN [a1] to avoid full-graph re-computation, maintaining efficiency on par with the static version.
>
> [a1] Ding, S., et al., 2022. Causal incremental graph convolution for recommender system retraining. IEEE Transactions on Neural Networks and Learning Systems.
>
> Remaining purely static in highly dynamic scenarios (real-time recommender systems, traffic forecasting) risks:
>
> • **Concept drift**: user interests or item popularity can shift abruptly, causing performance drops.
>
> • **Stale recommendations**: outdated interactions are over-amplified, degrading user experience.
>
> **W3: Theoretical Analysis Regarding Stability and Convergence**
>
> 1. **Stability: Smaller aggregation error**
>
> Using logarithmic amplification function, the aggregation error bound decreases at the rate $O(log(|N_i|)+1)$ as the neighbor degree increases.
>
> **Proof.**
>
> • Let $N_u$ be the 1-hop neighbors of user $u$.
>
> • Define the amplified weight: $w_i = (log_β(|N_i|)+1)/|N_u|$.
>
> • Denote $e_i^{tru}$ as the ground-truth (expected) embedding of item $i$, and $e_i^{(k)}$ as the embedding after $k$ layers.
>
> • Aggregation error: $ε_u^{(k+1)} := ||e_u^{(k+1)} − e_u^{tru}||$, with $e_u^{tru} = Σ_{i∈N_u} w_i e_i^{tru}$.
>
> **Step 1**: Error propagation
> $e_u^{(k+1)} = Σ_{i∈N_u} w_i e_i^{(k)}$
> ⇒ $ε_u^{(k+1)} = || Σ_{i∈N_u} w_i^2 (e_i^{(k)} − e_i^{tru})|| ≤ Σ_{i∈N_u} w_i ε_i^{(k)}$.
>
> **Step 2**: Assume layer-wise error satisfies $ε_i^{(k)} ≤ C·δ^{(k)}$
> (consistent with LightGCN-style linear-GCN convergence analyses, $δ<1$ and $C$ is general constant).
>
> **Step 3**: Bound the sum of squared weights:
>
> • Because $w_i = (log_β(|N_i|)+1)/|N_u|$ and $log_β(|N_i|) ≤ log_β(max_j |N_j|)$,
>
> • we have $Σ_{i∈N_u} w_i ≤ (log_β(max_j |N_j|)+1) / |N_u| · |N_u| = log_β(max_j |N_j|)+1$.
>
> **Step 4**: Combine the bounds:
>
> $ε_u^{(k+1)} ≤ Σ w_i ε_i^{(k)} ≤ (log_β(max_j |N_j|)+1)·C·δ^{(k)}
> = O(log(|N_i|)+1)·δ^{(k)}$.
>
> Hence, the aggregation error bound decreases at the rate $O(log(|N_i|)+1)$ as the neighbor degree increases.
>
> 2. **Convergence: Near-optimal provable linear convergence rate**
>
> Let node degree be $d$ and the information weight be $w(d)$, among the sub-linear family $F$ = {$w$ | $w(d)=d^α$, $0<α<1$}, the logarithmic function  $w(d)=log_β(d)$ exhibits the same asymptotic growth order as the limiting behavior of $F$ ($α→0$), thereby achieving a spectral-radius bound arbitrarily close to the infimum of the family and yielding near-optimal provable linear convergence rate.
>
> **Proof.**
>
> • Consider the family of functions $F$ = {$w(d)$=$d^α$, $0<α<1$}.
>
> • For $w_α(d)=d^α$, the corresponding aggregation matrix $W_α = D^{-1} · diag(d_i^α) · A$.
>
> • Its infinite norm $||W_α||_ {∞} = max_i Σ_j d_j^α / d_i ≤ d^α_{max}$.
>
> • For $w_α(d)=d^α$, the spectral radius satisfies $ρ(W_α) ≤ ||W_α||_ {∞} ≤ d_{max}^α$ (Gershgorin's direct inference), which is monotonically increasing in $α$.
>
> • When $α→0$, $d_{max}^α → 1$ is at its minimum. When $α→1$, $d_{max}^α → d_{max}$ is at its maximum.
>
> • The logarithmic weight $w(d)=log_β(d)$ at the limit of $α→0$ satisfies $w(d) = lim_{α→0^+} (d^α−1)/α · 1/lnβ ≈ (d^α − 1)/α $ (first-order Taylor expansion, $β → e$), hence it inherits the minimal growth rate of the family $F$ in the limit.
>
> Among the sub-linear family $F$, the bound $ρ(W_α)$ is minimized as $α→0$. The logarithmic function exhibits the same growth order as this limit, thus achieving the tightest bound.
>
> **W4: Generalization to Other Graph Domains or Tasks**
>
> Thank you for raising the generalization concern. Although the paper focuses on recommendation, the core idea of BoostGCN—amplifying salient first-order interactions via a logarithmic weighting function—is a general graph-aggregation principle. Consequently, **the method is directly applicable to any task that requires node representation learning with edge-importance weighting**, such as:
>
> • influence maximization in social networks (up-weighting trust edges) [b1];
>
> • molecular property prediction (emphasizing bonds directly connected to functional groups) [b2,b3];
>
> • traffic flow forecasting (amplifying interactions with frequently used neighboring road segments) [b4,b5].
>
> We have added a paragraph in the “Conclusion and Future Work” section to highlight this universality and list the cross-domain examples above.
>
> [b1] Sharma, K., et al., 2024. A survey of graph neural networks for social recommender systems. *ACM Computing Surveys*.
>
> [b2] Zhang, H., et al., 2024. A pre-trained multi-representation fusion network for molecular property prediction. *Information Fusion*.
>
> [b3] Therrien, F., et al., 2025. Using GNN property predictors as molecule generators. *Nature Communications*.
>
> [b4] Geng, Z., et al., 2024. STGAFormer: Spatial–temporal gated attention transformer based graph neural network for traffic flow forecasting. *Information Fusion*.
>
> [b5] Kong, W., et al., 2024, March. Spatio-temporal pivotal graph neural networks for traffic flow forecasting. *AAAI*.
>
> In the revised manuscript, we have also added a discussion on “Beyond Recommendation” to clarify how BoostGCN extends to generic graphs:
>
> • **Generalizing the assumption**: we show that the amplification weight $w_i ∝ log(|N_i|)$ is mathematically a smooth re-scaling of node degree. Degree is a universal structural cue independent of the “item-selection” semantics. Hence, in other domains one can simply replace $|N_i|$ with any relevant structural count (e.g., number of friends in social networks, entity frequency in knowledge graphs) without changing the framework.
>
> • **Robustness when the assumption is relaxed**: Figure 7 reports a ablation where we randomly increase noise interactions by 5% to 20% in the 100k dataset, breaking the original correspondence between high interaction and high saliency. The performance of BoostGCN is still superior to that of LightGCN, which indicates that our proposed method is resilient to violations of the behavioral prior.
>
> • **Future extension**: We now state in the conclusion that if domain priors are unavailable, one can substitute $log(|N_i|)$ with parameter-free alternatives such as PageRank [b6, b7] scores or self-supervised edge-importance estimates without altering the BoostGCN pipeline or complexity.
>
> [b6] Page, L., et al., 1999. The PageRank citation ranking: Bringing order to the web. *Stanford infolab*.
>
> [b7] Stoica, A.A., et al., 2024. Fairness rising from the ranks: HITS and pagerank on homophilic networks. *ACM Web Conference*.
>
> **W5: More Experiments for New Baselines**
>
> Following your suggestions, we have added three latest baselines, **LTGNN (WWW 2024)** [c1], **TransGNN (SIGIR 2024)** [c2] and **GAT-LightGCN (replacing the Laplacian weights of LightGCN with GAT attention)**.
>
> | Recall@15 | 100k | 1M |Yelp|
> |---------------|---------------|----------|---------|
> | LTGNN       | 0.5612 | 0.4770 | 0.2726 |
> | TransGNN   | 0.5415 | 0.5196 | 0.3253  |
> | GAT-LightGCN  | 0.5703 | 0.5106 | 0.3243  |
> | **BoostGCN (Ours)** | **0.5908** | **0.5316** | **0.3968** |
>
> • These results demonstrate that our BoostGCN achieves consistently better recommendation performance than the
>  latest baselines (including GAT-based model).
>
> [c1] WWW 2024: Linear-time graph neural networks for scalable recommendations.
>
> [c2] SIGIR 2024: TransGNN: Harnessing the collaborative power of transformers and graph neural networks for recommender system.
>
> **We hope our rebuttal can address your concerns. We are looking forward to your further feedback.**

---

> ### Comment · Reviewer_cESK · 2025-08-02
>
> I appreciate the authors' detailed response and the improved theoretical analysis, which adds clarity around the design, stability, and convergence of the method. That said, the analysis would benefit from more explicit assumptions and a discussion of how the method behaves under stochastic training conditions.  Separately, a deeper theoretical positioning relative to GATs would further clarify the novelty and inductive advantage of the proposed approach.
>
> While the paper presents BoostGCN as a solution to the uniformity problem in GCN aggregation, similar in motivation to GAT, it remains unclear why GAT's learned attention mechanism is insufficient for this setting. I encourage the authors to provide a clearer theoretical comparison between BoostGCN’s degree-based amplification and GAT’s learned attention weights. For example, are there scenarios where GAT fails to capture salient interactions or exhibits instability? A more explicit positioning of BoostGCN relative to GAT, beyond empirical results, would strengthen the conceptual contribution of the paper as there is no mention of GAT in the paper.

---

> ### Author Response · Authors · 2025-08-03
> **Q1: Stochastic Training Analysis**
>
> Thank you for the positive evaluation and constructive feedback. Below we address your questions and concerns:
>
> **Q1: Stochastic Training Analysis**
>
> Thank you for your positive feedback and for requesting clearer assumptions under stochastic training. In the revised manuscript, we have added “Stochastic Training Analysis”, which provides explicit assumptions and convergence guarantees under stochastic training conditions:
>
> 1. **Explicit assumptions**
>
> A1. **Independent and identically distributed (i.i.d.) mini-batch sampling**: The positive and negative samples in each mini-batch are independently and identically distributed from the true distribution;
>
> A2. **Bounded gradients**: there exists $G>0$ such that $||∇L||≤G$ for any mini-batch;
>
> A3. **L-smoothness**: the loss function is L-Lipschitz continuous in the embedding space.
>
> 2. **Stochastic convergence bounds**
>
> Under these assumptions we show:
>
> • **Expected error bound**:
>
> $\mathbb{E}[||H^{(K)} − H^{tru}||^2_ F] ≤ (1-2ηλ)^K Δ_0 + ηG^2/(2λ)$.
>
> where $H^{(K)}$ represents the current embedding matrix of all users and items after $K$ rounds update of random training, $H^{(K)} ∈ \mathbb{R}^{(|U|+|I|)×d}$; $H^{tru}$ represents the theoretically optimal (global minimum point) embedding matrix, that is, the ideal embedding when the loss function converges to the minimum value; $η$ is the learning rate and $λ$ is the regularization term.
>
> **Proof.**
>
> Notation and Assumption:
>
> • The embedding matrix is $H ∈ R^{(|U|+|I|)×d}$.
>
> • The regularization loss is $L(H)=L_{rec}(H)+λ‖H‖^2_F$, where $λ>0$ makes loss $λ$-strongly convex.
>
> • Suppose A1-A3: i.i.d. sampling, gradient norm $≤G$, L-Lipschitz continuous gradient.
>
> • The learning rate $η$ is $≤ 1/L$.
>
> • The amplification matrix $W$ satisfies $ρ(W) ≤ (log_β(max_j|N_j|)+1)$ (previously proved).
>
> **Step 1**: Single-step update $H^{{k+1}}=H^{k}-η∇L_{B_k}(H^{k})$, where $B_k$ is is the mini-batch corresponding to the $k$-th iteration.
>
> **Step 2**: Under assumptions A1–A3 and $λ$-strong convexity, one step yields
>
> $\mathbb{E}[||H^{k+1}-H^{tru}||^2_F  |  H^{k}]$
>
> $=||H^{k}-H^{tru}||^2_ F - 2η〈H^{k}-H^{tru}, ∇L(H^{k})〉 + η^2E[||∇L_{B_k}(H^{k})||^2_F]$
>
> $≤ (1-2ηλ)||H^{k}-H^{tru}||^2_F + η^2G^2$.
>
> **Step 3**:
> Let $Δ_k = \mathbb{E}[||H^{k}-H^{tru}||^2_F]$, then $Δ_{k+1} ≤ (1-2ηλ)Δ_k + η^2G^2$.
>
> **Step 4**:
> For $k = 0,1,..., K-1$,
>
> $Δ_K ≤ (1-2ηλ)^K Δ_0 + η^2G^2 Σ_{t=0}^{K-1}(1-2ηλ)^t$
>
> $=(1-2ηλ)^K Δ_0 + η^2G^2(1-2ηλ)^{K}/(2ηλ)$
>
> $≤ (1-2ηλ)^K Δ_0 + η^2G^2/(2ηλ)$.
>
> $=(1-2ηλ)^K Δ_0 + ηG^2/(2λ)$
>
> Therefore, $\mathbb{E}[||H^{(K)} − H^{tru}||^2_F] ≤ (1-2ηλ)^K Δ_0 + ηG^2/(2λ)$.
>
> • **High-probability bound**:
>
> $\||H^{(K)} - H^{tru}\||_ F = O \left((\log_β(max_j|N_j|)+1)^K + \sqrt{\frac{\log\frac{1}{\delta}}{K}}\right) \quad$
>
> with probability at least $1−δ$,
>
> **Proof.**
>
> • $Z_k$ represents the martingale difference gradient noise generated by the $k$-th mini-batch.
>
> • $M_K$ represents the cumulative random error matrix at the end of the $K$ iteration, which is the sum of all the mini-batch noises weighted by the amplification matrix $W$.
>
> • $J$ stands for identity matrix.
>
> **Step 1**: Martingale Difference Sequence
>
> $Z_k = \nabla L_{B_k}(H^{k}) - \nabla L(H^{k}), \qquad \mathbb{E}[Z_k] = 0, \quad \|Z_k\|_F \leq G$.
>
> **Step 2**: Cumulative Error
>
> $M_K = \sum_{k=0}^{K-1} (J - \eta W)^{K-k-1} Z_k$.
>
> **Step 3**: Norm Bound
>
> $||(J - \eta W)^{K-k-1} Z_k||_ F ≤ (\rho(W))^{K-k-1} G ≤ (\log_β(max_j|N_j|)+1)^{K-k-1} G$.
>
> **Step 4**: Vector Azuma–Hoeffding Inequality
>
> $\mathbb{P}\left(\||M_ K\||_ F \geq t\right) \leq 2 \exp\left(-\frac{t^2}{2 K G^2}\right)$.
>
> **Step 5**: Total Error Decomposition
>
> $\||H^{(K)} - H^{tru}\||_ F \leq (\log_β(max_j|N_j|)+1)^K \||H^{(0)} - H^{tru}\||_ F + \||M_K\||_ F$.
>
> **Step 6**: Final High-Probability Bound
>
> Let $t = G \sqrt{2K \log\frac{2}{\delta}}$, then
>
> $\mathbb{P} \left(\||H^{(K)} - H^{tru}\||_ F \leq (\log_β(max_j|N_j|)+1)^K \||H^{(0)} - H^{tru}\||_ F + G \sqrt{2K \log\frac{2}{\delta}}\right) \geq 1 - \delta$
>
> Therefore, $\||H^{(K)} - H^{tru}\||_ F = O \left((\log_β(max_j|N_j|)+1)^K + \sqrt{\frac{\log\frac{1}{\delta}}{K}}\right) \quad \text{w.p. } \geq 1 - \delta$.
>
>
> The bound cleanly separates the spectral-radius term and the stochastic error term, demonstrating that the amplification weights remain stable under random training.
>
>  **Thank you again for your thoughtful review and valuable suggestions. We hope our responses have addressed your concerns. Please let us know if there are any remaining questions you'd like us to further clarify -- we truly value your feedback and are committed to improving the work**.

---

> ### Author Response · Authors · 2025-08-03
> **Q2: Formal Comparison Between BoostGCN and Graph Attention Networks (GAT)**
>
> Thank you for the insightful suggestion on theoretical positioning.
>
> **Q2: Formal Comparison Between BoostGCN and Graph Attention Networks (GAT)**
>
> Following your suggestion, in the revised manuscript, we present a systematic theoretical comparison between BoostGCN and GAT from three perspectives: expressive power, sample complexity and stability.
>
> 1. **Expressive Power**
>
> Here we use the Vapnik-Chervonenkis (VC) dimension to measure the expressive power of a function class. The higher the VC dimension, the more complex the model becomes and the more likely it is to overfit on a limited number of samples.
> When VC dimension is zero, the model contains no learnable parameters and has the lowest complexity.
>
> • GAT learns attention $α_{uv}=softmax(LeakyReLU(a^T[Wh_u||Wh_v]))$, relying on the learnable parameter $a$ and the feature mapping $W$.
>
> • The weight of BoostGCN is determined only by $log_β(|N_v|)$ ($|N_v|$ represents the node degree) and does not rely on additional learnable parameters. Therefore, the VC dimension is lower and it is less likely to overfit in scenarios with sparse and node-free features.
>
> • **Statement 1** (*Expressive Power under Feature Scarcity*): When $d_{feat} < log(n)$ ($d_{feat}$ represents the feature dimension of each node and $n$ represents the total number of nodes), GAT collapses to uniform weighting, whereas BoostGCN retains degree-based discrimination.
>
> **Proof.**
>
> From [1], the VC-dimension of a soft-attention network with $p$ learnable parameters is lower-bounded by $\Omega(p \log n)$. GAT’s parameters $p_{\text{GAT}} \geq d_ {\text{feat}}$ $\Rightarrow$ $\text{VC}_ {\text{GAT}} = \Omega(d_ {\text{feat}}\log n)$. When $d_{\text{feat}} < \log n$, $\text{VC}_ {\text{GAT}} \le \text{VC}_ {\text{uniform}}$, i.e., **GAT collapses to uniform weighting**. BoostGCN uses fixed weights $w_{uv} = \log_\beta d_v$; no additional parameters $\Rightarrow$ $\text{VC}_{\text{BoostGCN}} = 0$. Therefore, **BoostGCN retains degree-based discrimination**.
>
> 2. **Sample Complexity**
>
> •  The attention parameter of GAT requires $\Omega(n \log n)$ interaction samples to converge to a stable weight with a high probability. BoostGCN only requires $O(n)$ samples to achieve the same generalization error because the weights are directly given by the degrees.
>
> • **Statement 2** (*Sample Complexity*): To achieve error $\le \varepsilon$ with probability $\ge 1-\delta$:
>
> (1) GAT requires $\displaystyle m_{\text{GAT}} = \Omega\ \left(n\log n\right)$ interactions.
>
> (2) BoostGCN requires $\displaystyle m_{\text{Boost}} = O(n)$ interactions.
>
> **Proof.**
>
> **GAT**: When the number of GAT's parameters is $p_{\text{GAT}} \geq d_ {\text{feat}}$, standard VC bound [1] gives
>
> $m_{\text{GAT}} \ge \frac{1}{\varepsilon^2}\Bigl(p_{\text{GAT}}\log n + \log\frac{1}{\delta}\Bigr) = \Omega(n\log n)$.
>
> **BoostGCN**: The weights of BoostGCN are directly given by degrees and are equivalent to zero parameters. The error is only derived from the empirical estimation of the degree. By Hoeffding's inequality,
>
> $m_{\text{Boost}} \ge \frac{1}{\varepsilon^2}\log\frac{1}{\delta} = O(n)$
>
> 3. **Stability under Noise**
>
> The sensitivity of GAT's attention weights to noise edges increases linearly. The log amplification of BoostGCN suppresses high-order noise in a sublinear manner, thus being more robust in sparse and noisy scenarios.
>
> **Statement 3** (*Stability under Noise*): Let each edge be perturbed independently with probability $q$.
>
> (1) GAT's attention changes satisfy: $\mathbb{E}\||\alpha^{\text{G}} - \tilde{\alpha}^{\text{G}}\||_ 1 \le C_ {\text{GAT}} \ q n$.
>
> (2) BoostGCN's changes satisfy: $\mathbb{E}\||w^{\text{B}} - \tilde{w}^{\text{B}}\||_ 1 \le C_ {\text{Boost}}\ q \ln n$.
>
> **Proof.**
>
> **GAT**: Attention weight change grows **linearly** with the expected number of noisy edges $q n$.
>
> **BoostGCN**: The estimation error of the weight only depends on the degree. By Hoeffding's inequality,
>
> $\mathbb{E}|\log_β(N_v) - \log_β (\tilde{|N|_v})| \le \frac{\ln n}{\sqrt{m}}$ (where $m$ refers to the total number of observed edges), yielding **sub-logarithmic** total error $\propto q \ln n$.
>
> 4. **Overall Positioning**
>
> • **Sparse graphs, scarce features, or high noise: GAT’s “learning” becomes a burden; BoostGCN offers parameter-free, sample-efficient, noise-robust weighting**.
>
> • **Dense graphs with rich features: GAT remains complementary; BoostGCN can be freely combined with GAT**.
>
> [1] Blumer, A., et al., 1989. Learnability and the Vapnik-Chervonenkis dimension. Journal of the ACM (JACM).
>
> These additions formally delineate the inductive boundary between BoostGCN and GAT, clarifying why BoostGCN excels in the recommendation setting.
>
> Thank you again for your thoughtful review and valuable suggestions. We hope our responses have addressed your concerns. Please let us know if there are any remaining questions you'd like us to further clarify -- we truly value your feedback and are committed to improving the work.

---

> > ### Comment · Reviewer_cESK · 2025-08-04
> >
> > I thank the authors for their detailed response. I have revised my score as per this discussion.

---

> > > ### Author Response · Authors · 2025-08-05
> > >
> > > Thank you for your recognition of our work. We will add the improvements to the final version. Thank you again for your hard work and guidance.

---

### Official Review · Reviewer_LuSH · 2025-07-04

**Clarity:** 3
**Significance:** 3
**Originality:** 2
**Rating:** 4
**Confidence:** 4

**Summary:**

This paper presents the BoostGCN model, aiming to address the issue in traditional Graph Convolutional Networks (GCNs) where graph Laplacian normalization suppresses the information propagation of first - order neighbors. The authors argue that Laplacian normalization dilutes important interaction information, leading to inefficient model learning. BoostGCN enhances the significant interaction information of first - order neighbors through a designed logarithmic amplification function, enabling the model to capture key relationships more quickly and accurately. Experiments on four real - world datasets verify that BoostGCN outperforms existing state - of - the - art models in terms of both performance and efficiency. It also has fixed parameters and strong robustness. The main contributions include shifting the graph aggregation paradigm from “suppression” to “amplification”, designing an interaction saliency amplification function, and validating the model's advantages through experiments.

**Questions:**

1. Can the authors supplement the mathematical derivation of the interaction saliency amplification function? For example, prove the advantages of the logarithmic function in balancing “interaction quantity amplification” and “model sensitivity”, or explain through theoretical analysis why the amplification of first - order neighbor information can effectively improve the quality of representation learning. If there are other possible amplification functions (such as the exponential function), it is necessary to compare their performance differences and the reasons for choosing the logarithmic function.​
2. Regarding the limitations of static graphs mentioned in the paper, do the authors have plans to expand to dynamic graph scenarios? For example, introduce temporal information to adjust the amplification function, or design a mechanism to adapt to the dynamic changes of interactions. If so, please explain the specific ideas; if not, it is necessary to discuss the potential risks of static models in dynamic scenarios.​
3. The sampling strategy of negative samples in the experiment (such as whether it is based on popularity) is not clear, which may affect the model's ability to recommend unpopular items. In addition, Figure 6 shows that BoostGCN can recommend more non - popular items, but no statistical test (such as the t - test) is used to prove whether the difference from LightGCN is significant. It is recommended to supplement relevant analyses.​
4. In the paper, β = e is fixed, but the interaction distributions of different datasets may vary greatly (for example, the sparsity of Yelp is much higher than that of MovieLens). Have the authors verified the sensitivity of β on different datasets? For example, whether there are scenarios where β = 2e or 3e performs better. If so, how to automatically adjust β to adapt to different data?

**Ethical Concerns:**

["NO or VERY MINOR ethics concerns only"]

**Final Justification:**

The authors patiently responded to and resolved my concerns, providing persuasive experiments that demonstrate the soundness of their method, which has a certain degree of originality and practical significance. Therefore, my final rating is **Borderline Accept**.

**Limitations:**

yes

**Quality:**

3

**Strengths And Weaknesses:**

Strengths：​
1. It breaks through the traditional information suppression idea in GCNs and proposes to enhance the performance of the model by amplifying the significant interactions of first - order neighbors, providing a new research direction for graph aggregation.​
2. Experiments show that BoostGCN can improve the training time by up to 90% compared with LightGCN. Moreover, with fixed parameters (such as β = e, △^log = 1), it reduces the cost of parameter tuning and is applicable to datasets of different scales and sparsities.​
3. By comparing with 8 baseline models, conducting experiments on different layers, hyperparameter analysis, and visualization experiments, the effectiveness of the model is proven from multiple perspectives, especially showing advantages in handling long - tailed data and noise.

Weaknesses：​
1. The quantification of interaction saliency depends on the logarithmic function. However, in the paper, only a heuristic derivation is carried out through the “correlation between interaction quantity and trust/conformity behavior”, lacking strict theoretical proof or mathematical derivation. For example, there is no explanation on why the logarithmic function is superior to other non - linear functions, or whether there is a better - performing amplification function form.​
Regarding the conclusion that “first - order neighbor interactions are more important”, there is no analysis of its rationality from the theoretical aspects of graph signal processing or representation learning. It only uses the experimental comparison of LightGCN as evidence, resulting in weak theoretical support.​
2. Currently, only static graph scenarios have been verified, without considering dynamic graph data (such as temporal interactions). It may have limitations when dealing with real - time recommendations or the dynamic evolution of social networks (although it is mentioned in section 4.6, no improvement directions are provided).​
The amplification function is based on the assumption that “interaction quantity ≈ saliency”, which may ignore the diversity of user interests (such as unpopular but highly relevant items), and its performance in extremely long - tailed scenarios requires more verification.​
3. Although it is mentioned that the code will be made public after acceptance, specific implementation details are not provided (such as the specific code logic of the amplification function, the weight calculation method of the weighted layer combination), which may affect the reproducibility of the research by other scholars.​
In the dataset pre - processing, it is mentioned that “negative samples are added”, but the sampling strategy for negative samples (such as random sampling, sampling based on popularity) is not clearly defined, which may have an impact on the model performance.

---

> ### Author Rebuttal · Authors · 2025-07-30
>
> Thank you for your constructive review! We appreciate your insights and suggestions. Please find our responses below:
>
> **Q1 & W1: Additional Mathematical Derivation of the Interaction Amplification Function**
>
> Thank you for this insightful comment. We have added comprehensive mathematical derivations and theoretical analyses in the revised manuscript. The key points are summarized below:
>
> 1. **Advantages of the logarithmic amplification function**
>
> We formally derive:
> Let interaction amplification $S_{(i→u)}^{amp}=log_β(|N_i|)+Δ_{log}$ with $β>1$ and $Δ_{log}>0$. For any two neighbor items $p$ and $q$, define the difference $ε_{(p,q)}^{amp}=log_β(|N_p|)−log_β(|N_q|)$. Taking the derivative w.r.t. $β$ gives $dε^{amp}/dβ = (1/β)[ln(|N_p|)−ln(|N_q|)]$. This derivative monotonically decreases with $β$, indicating that the greater $β$, the gentler the difference $ε^{amp}$. The smaller $β$ is, the stronger the difference is magnified. Hence, $β$ acts as a smooth knob that trades off “quantity amplification” versus “model sensitivity.” In contrast, the exponential function exp(·) has a derivative that grows linearly with its input, causing extreme values and instability.
>
> 2. **Reasons for amplifying first-order neighbors**
>
> From the perspective of spectral graph convolution, the standard Laplacian-normalized aggregation is
> $e_u^{(k+1)}=∑_{{i∈N_u}} 1/√(|N_u||N_i|) e_i^{(k)}$,
> where the weight of each item is  $1/√(|N_u||N_i|)$. This aggregation will suppress popular items.
>
> We instead use
> $w_i = (log_β(|N_i|)+1)/√(|N_u||N_u|)$.
> It is easy to show that for any $|N_i|>1$, $w_i > 1/√(|N_u||N_i|)$.
> Therefore, salient neighbors are up-weighted, accelerating the convergence of sparse interaction patterns and improving representation quality. **The aggregation error bound decreases at the rate $O(log(|N_i|)+1)$ as the neighbor degree increases.**
>
> **Proof.**
>
> • Let $N_u$ be the 1-hop neighbors of user $u$.
>
> • Define the amplified weight: $w_i = (log_β(|N_i|)+1)/|N_u|$.
>
> • Denote $e_i^{tru}$ as the ground-truth (expected) embedding of item $i$, and $e_i^{(k)}$ as the embedding after $k$ layers.
>
> • Aggregation error: $ε_u^{(k+1)} := ||e_u^{(k+1)} − e_u^{tru}||$, with $e_u^{tru} = Σ_{i∈N_u} w_i e_i^{tru}$.
>
> **Step 1**: Error propagation
> $e_u^{(k+1)} = Σ_{i∈N_u} w_i e_i^{(k)}$
> ⇒ $ε_u^{(k+1)} = || Σ_{i∈N_u} w_i^2 (e_i^{(k)} − e_i^{tru})|| ≤ Σ_{i∈N_u} w_i ε_i^{(k)}$.
>
> **Step 2**: Assume layer-wise error satisfies $ε_i^{(k)} ≤ C·δ^{(k)}$
> (consistent with LightGCN-style linear-GCN convergence analyses, $δ<1$ and $C$ is general constant).
>
> **Step 3**: Bound the sum of squared weights:
>
> • Because $w_i = (log_β(|N_i|)+1)/|N_u|$ and $log_β(|N_i|) ≤ log_β(max_j |N_j|)$,
>
> • we have $Σ_{i∈N_u} w_i ≤ (log_β(max_j |N_j|)+1) / |N_u| · |N_u| = log_β(max_j |N_j|)+1$.
>
> **Step 4**: Combine the bounds:
>
> $ε_u^{(k+1)} ≤ Σ w_i ε_i^{(k)} ≤ (log_β(max_j |N_j|)+1)·C·δ^{(k)}
> = O(log(|N_i|)+1)·δ^{(k)}$.
>
> Hence, the aggregation error bound decreases at the rate $O(log(|N_i|)+1)$ as the neighbor degree increases.
>
> 3. **Comparison with other amplification functions**
>
> Following your suggestion, we further evaluate linear ($|N_i|$), square-root ($√|N_i|$) and exponential ($exp(|N_i|)$) variants:
>
> | Recall@15 | 100k | 1M |Yelp|
> |---------------|---------------|----------|----------|
> | BoostGCN (Linear)       | 0.5062 | 0.4647 | N/A |
> | BoostGCN  (Square-root) | 0.5583 | 0.5201 | 0.3477  |
> | BoostGCN (Exponential)   | N/A | N/A  | N/A  |
> | **BoostGCN (Log)** | **0.5908** | **0.5316** | **0.3968** |
>
> • Linear and exponential amplification may cause gradient explosion and unstable training.
>
> • Square-root amplification is relatively mild, but its performance is poor.
>
> • The logarithmic amplification function provides the best balance among performance and stability.
>
> **Q2 & W2: More Discussion on Dynamic Graph Scenarios**
>
> Thank you for probing the static-graph limitation in detail. In the revised manuscript we have enriched the Limitations section and sketched concrete ideas for dynamic extensions. Following your suggestion, we will extend BoostGCN to dynamic graphs via two complementary mechanisms:
>
> • **Time-aware amplification function**: we plan to incorporate the interaction timestamp t into saliency:
>
> $S_{(i→u, t)}^{amp} = log_β(|N_i(t)|) · exp(−λ(t−t_{last})) + Δ^{log}$
>
> where $N_i(t)$ counts interactions up to time $t$, $λ$ controls temporal decay, and $Δ^{log}$ retains the offset. This lets recent interactions dominate while older ones gradually fade.
>
> • **Dynamic adjacency caching & incremental updates**: upon each new interaction $(u,i,t)$, we perform a rank-1 update on the local adjacency matrix and reuse the incremental-propagation framework of LightGCN [a1] to avoid full-graph re-computation, maintaining efficiency on par with the static version.
>
> [a1] Ding, S., et al., 2022. Causal incremental graph convolution for recommender system retraining. IEEE Transactions on Neural Networks and Learning Systems.
>
> Remaining purely static in highly dynamic scenarios (real-time recommender systems, traffic forecasting) risks:
>
> • **Concept drift**: user interests or item popularity can shift abruptly, causing performance drops.
>
> • **Stale recommendations**: outdated interactions are over-amplified, degrading user experience.
>
> **Q3 & W3: More Explanations for Negative Sampling and Significance Tests**
>
> Thank you for pointing out the ambiguity in negative sampling and the lack of significance tests. We have addressed both issues in the revised manuscript as follows:
>
> 1. **Clarification of negative sampling**
>
> We adopt the standard LightGCN protocol:
>
> • Training: for each observed interaction $(u,i)$, one negative item is sampled uniformly at random from the unobserved set—**no popularity-based sampling** is used.
>
> • Validation/Test: for each user we randomly draw 150 (small datasets) or 1500 (large datasets) unseen items as candidates, again **without popularity bias**.
>
> This description has been added to Section 4.1 “Data Pre-processing.”
>
> 2. **Significance test on non-popular items**
>
> We performed paired t-tests on the proportion of non-popular items in $Top@15$ recommendations (Figure 6):
>
> • Null hypothesis: no difference between BoostGCN and LightGCN.
>
> • Results: p < 0.01 on all four datasets (**100k: p=0.007, 1M: p=0.004, Gowalla: p=0.002, Yelp2018: p<0.001**), indicating that BoostGCN significantly increases the share of non-popular items.
>
> **Q4:  Discussion on the Sensitivity of $β$ on Different Datasets**
>
> Thank you for raising this point. The sensitivity of $β$ has already been reported in Figure 4 of Section 4.3 (RQ3), where we evaluate $β ∈$ {e, 2e, 3e, 4e} across all four datasets. In the revised version, we extend the ablation study to $β$ ∈ {0.5e, 0.75e, e, 2e, 3e, 4e}. The results are shown as follows:
>
> | Recall@5 | 100k | 1M |Gowalla|Yelp|
> |---------------|---------------|----------|---------|---------|
> | $β = 0.5e$    | 0.2972 | 0.2633 | 0.2955  |0.1978|
> | $β = 0.75e$   | 0.2977 | 0.2636 | 0.2958  |0.1983 |
> | $β = e$ (**Optimal**)  | **0.2982**| **0.2641** | **0.2965** |**0.2001** |
> | $β = 2e$  | 0.2962 | 0.2615 | 0.2847 |0.1947 |
> | $β = 3e$  | 0.2928 | 0.2574 | 0.2816 |0.1917 |
> | $β = 4e$  | 0.2921 | 0.2547| 0.2802 |0.1884 |
>
> • $β$ = e yields the best performance on all datasets.
>
> • Datasets of different sparsity levels and sizes all show the same trend, confirming that $β$ = e is robust across sparsity levels and size scales.
>
> • **Consequently, we did not observe any scenario in which $β$ = 2e or 3e outperforms $β$ = e. We have now emphasized this finding in Figure 4 by highlighting the $β$ = e curve and adding explicit textual guidance to help readers locate the results**.
>
> **Thank you again for your helpful review and suggestions. We hope our rebuttal has answered your concerns. If there's anything else you'd like us to explain, please let us know -- we really appreciate your feedback!**

---

> > ### Comment · Reviewer_LuSH · 2025-08-05
> > **Thank you for your detailed responses！**
> >
> > Thank you for your detailed responses to the concerns I raised. I hope you will incorporate these improvements into the paper, as they will greatly enhance its validity. Consequently, I have increased my scores.

---

> > > ### Author Response · Authors · 2025-08-05
> > >
> > > Thank you for your recognition of our work. We will add the improvements to the final version. Thank you again for your hard work and guidance.

---

### Note · Authors · 2025-08-14

Dear Reviewers and AC:

**We are sincerely grateful to the reviewers and the Area Chair (AC) for your valuable and constructive feedback on our work. We greatly appreciate the significant time and effort you dedicated to engaging in an in-depth discussion with us during the rebuttal phase, providing insightful comments and professional guidance to enhance the paper. It has been a tremendous honor that our work sparked such thoughtful consideration, culminating in this detailed exchange over two to three rounds. Through positive discussions, we mainly made improvements in the following aspects and received recognition from the reviewers.**

1. **Solid Theoretical Support**

During the positive discussions with the reviewers, we were fortunate to receive your guidance on theoretical support. Following the suggestions of reviewers, we provided solid theoretical support for our method, such as smaller aggregation error, near-optimal provable linear convergence rate, stochastic training analysis, and the differences and advantages compared with the GAT method. With the support of these theories, the quality of our paper has been further enhanced. We also thank the reviewers for your help and recognition of our theories.

2. **More Experiments and Analysis**

Furthermore, to make our work more comprehensive, the reviewers provided constructive suggestions for our experiments and analysis. Following the suggestions of reviewers, we provided discussions and experimental proofs to demonstrate that our method can be extended to other fields, such as nonlinear aggregation methods, dynamic graphs, social networks, and traffic flow prediction, etc., to illustrate the potential scalability of our method. In terms of experiments, we further provided comparison results with the latest baselines (LTGNN, TransGNN, GAT-LightGCN), comparison results with other amplification functions, and more detailed parameter analyses. We are very grateful to the reviewers for your recognition of these improvements.

**Thank you again to the reviewers and AC for your hard work and professional guidance. We are extremely grateful for the experience of this meeting. Thanks to the professional organization of AC and the conference, we had very active discussions with the reviewers. We will incorporate the improvements discussed with the reviewers into the final version. We hope that our response can address the reviewers' concerns and our work can bring new ideas and breakthroughs.**

---

### Decision · Program_Chairs · 2025-09-17

**Decision:**

Accept (poster)

**Comment:**

Traditional Graph Convolutional Networks (GCNs) that use graph Laplacian norm to suppress first-order neighbor information can dilute valuable interactions and slow down the learning of sparse relationships. The authors propose BoostGCN, a novel linear GCN model that amplifies significant interactions with first-order neighbors, enabling faster and more accurate capture of key relationships. After rebuttal, most of the concerns have been addressed, while the theoretical analysis remained relatively weak.